# Heterogeneously deacetylated chitosans possess an unexpected regular pattern favoring acetylation at every third position

Margareta J. Hellmann [1], Dominique Gillet[2], Stéphane Trombotto[3], Sonja Raetz [1], Bruno M. Moerschbacher [1] ✉ & Stefan Cord-Landwehr [1]

Chitosans are promising biopolymers for diverse applications, with material properties and bioactivities depending *i.a.* on their pattern of acetylation (PA). Commercial chitosans are typically produced by heterogeneous deacetylation of chitin, but whether this process yields chitosans with a random or block-wise PA has been debated for decades. Using a combination of recently developed in vitro assays and in silico modeling surprisingly revealed that both hypotheses are wrong; instead, we found a more regular PA in heterogeneously deacetylated chitosans, with acetylated units overrepresented at every third position in the polymer chain. Compared to random-PA chitosans produced by homogeneous deacetylation of chitin or chemical *N*-acetylation of polyglucosamine, this regular PA increases the elicitation activity in plants, and generates different product profiles and distributions after enzymatic and chemical cleavage. A regular PA may be beneficial for some applications but detrimental for others, stressing the relevance of the production process for product development.

Chitosans are versatile functional biopolymers with superior material properties and diverse biological activities, offering huge potential for the growing bioeconomy. These linear, polycationic polysaccharides consist of β-1,4-linked *N*-acetyl-D-glucosamine units (GlcNAc, A-units) and D-glucosamine units (GlcN, D-units), thus differing in their degree of polymerization (DP), fraction of acetylation (FA), and pattern of acetylation (PA)[1]. Chitosans with these differing structural characteristics possess diverse biological activities such as antimicrobial properties[2] and the ability to interact with cell surfaces and specific receptors[3–5]. These activities are exploited in applications such as agriculture[6,7], the food and feed industry[2], water purification[8], cosmetics[9], and (veterinary) medicine, where they are used as wound dressings[10] or to prepare nanoparticles[11] as vehicles for drug or vaccine delivery.

Currently, commercial chitosans are derived from chitin, a homopolymer of β-1,4-linked GlcNAc units found abundantly in waste materials such as the exoskeletons of crustaceans and insects, the endoskeletons of cephalopods, and the cell walls of fungi[12]. There are several ways to produce chitosans from chitin. First, chitin can be partially deacetylated in an alkaline solution, typically under heterogeneous deacetylation (HTDA) conditions at high temperature, with short reaction time, and reactants in different phases[13] (insoluble crystalline chitin in a highly concentrated solution of NaOH)[14–16], but also under (what has come to be known as) homogeneous deacetylation (HMDA) conditions (lower concentration of NaOH, moderate temperature, and longer reaction time, so that the chitin crystals can swell faster than becoming deacetylated, emulating a truly monophasic and hence homogeneous[17] deacetylation reaction)[18,19]. Second, chitin can be completely deacetylated to polyglucosamine, and then treated by partial chemical *N*-acetylation (CNA) to produce chitosans[20,21]. It should be noted that chitin has two major crystalline polymorphic forms (α and β) that differ in the orientation of adjacent

[1]Institute for Biology and Biotechnology of Plants, University of Münster, 48143 Münster, Germany. [2]Gillet Chitosan SAS, La Ville Es Comte, 22350 Plumaudan, France. [3]Ingénierie des Matériaux Polymères (IMP), UMR 5223, Université Claude Bernard Lyon 1, CNRS, INSA Lyon, Université Jean Monnet Saint-Etienne, F-69622 Villeurbanne, France. ✉e-mail: moersch@uni-muenster.de

chitin sheets[22]. The more prominent α-chitin is found in crustacean shells, insect exoskeletons and fungal cell walls, whereas the endoskeleton of squids consists of β-chitin[23]. Because β-chitin exhibits weaker self-association through intermolecular interactions than α-chitin, the former swells faster in alkaline solutions and is therefore deacetylated more rapidly and efficiently[15].

The widespread application of chitosans as multifunctional biopolymers has been limited by the poor reproducibility of observed effects, preventing the development of products with reliable performance. This mainly reflects the poor structural characterization of chitosans, given that the DP, FA and PA can all have a profound effect on the physicochemical properties[21] and bioactivities[1] of these molecules. The full exploitation of chitosans in the bioeconomy depends on an in-depth understanding of their structure-function relationships. Standard methods are available to determine the DP[20] and FA[24,25], and second-generation chitosans with a defined DP and FA are now reaching the market. But analyzing the PA is more challenging. The current gold standard is the quantification of AA, DD, AD and DA diads by nuclear magnetic resonance (NMR) spectroscopy[14,26–28], which results in PA values ranging from 0 (perfectly block-wise) through 1 (perfectly random) to 2 (perfectly alternating)[27,29]. Less accurate methods include the analysis of DP distributions after chemical degradation with nitrous acid[30,31], or considerations based on solubility, crystallinity, enzymatic degradability, or behavior in solution or during dissolution[15,19,32,33]. The research community agrees that the very rare commercial HMDA and CNA chitosans have a random PA[14,19,26,27,31–33], but there is no consensus on the PA of HTDA chitosans, which make up nearly all commercial chitosan products. There is some evidence for a block-wise PA[19,32–34], but most authors argue in favor of random PA[14,26,27,30,35], and some suggest that the deacetylation occurs first block-wise on the crystalline substrate and then continues in a random manner once the material becomes amorphous[15,31,36].

Enzymatic digestion followed by mass spectrometry (MS) is the most recent method for the fingerprinting of chitosans[37], and is based on the unique subsite preference of a chitinosanase from *Alternaria alternata* (AaChio)[38]. Whereas chitinases require an A-unit bound at subsite −1 (GH18 chitinases) or subsite +1 (GH19 chitinases)[39–41], and most chitosanases require a D-unit at subsite −2 or at subsites −2 and −1[42], AaChio is absolutely specific for D-units at subsite −2 and for A-units at subsite −1, but lacks specificity at subsites +1 and +2. This enzyme therefore hydrolyzes chitosans precisely after each A-unit that is preceded by a D-unit. As described previously[37,38] and in detail in the supplementary information (Supplementary Fig. 4, Supplementary Eq. (1) and Eq. (2)), AaChio-MS fingerprinting allows us to calculate the average A- and D-block sizes.

Here we combined AaChio-MS fingerprinting[37] and size exclusion chromatography coupled to refractive index and MS detection (SEC-RI-MS) for product analysis[43] to screen a large set of chitosans produced by different methods (HTDA, HMDA and CNA), aiming to answer the decades-old and still unresolved question: what is the PA of HTDA chitosans? Surprisingly, in vitro experiments combined with in silico modeling of chitosan populations and their enzymatic cleavage products clearly indicated that HTDA chitosans possess neither a random nor block-wise PA, but instead a more regular PA. We show that this influences the product profiles of hydrolytic enzymes as well as the bioactivity of chitosans in plants, and we propose a hypothesis to explain how this PA is formed during deacetylation. Given that commercially available chitosans are almost exclusively produced by HTDA, our findings are highly relevant when aiming to determine the structure-function relationships of these products to optimize their performance. Our data also suggest that commercial chitosans fundamentally differ in their structure from chitosans which can be extracted from e.g. Mucoromycetes cell walls[44,45] where they are produced by the in vivo action of chitin deacetylases[46] on the nascent chitin chains before they form crystalline fibers. It is highly unlikely,

that these enzymes generate the exact same, regular PA as generated by HTDA.

## Results

### Heterogeneously deacetylated chitosans differ from homogeneously deacetylated and chemically N-acetylated chitosans

Initially, we compared HTDA, HMDA and CNA chitosans with the same FA by hydrolysis using AaChio followed by SEC-RI-MS analysis of the products. We chose chitosans with an intermediate FA of 0.32 because low FA samples may not possess enough A-units to identify patterns, and high FA HTDA samples are neither common nor easily available. Also, such chitosans can be depolymerized by both chitinases and chitosanases, but to a limited extent only, so that intermediate size oligomers are produced. Unexpectedly, very clear differences were observed in the RI signals generated by the oligomeric products of the enzymatic digestion when separated by DP (Fig. 1a). The products of the HMDA and CNA chitosans showed a very similar uniform DP distribution, with the highest proportion of products covering the range DP 3–5, whereas the DP distribution of the HTDA chitosan products was strikingly different, with DP 3, 6, 9 and 12 detected in unexpectedly large amounts. To quantify the overrepresentation of these DP = 3n products, we introduced a parameter known as triad strength (Fig. 1b). This is calculated using Eq. (1), where $I_{DPx}$ is the integrated RI signal of DP x (baseline from RI minimum between peaks for DP 6/7 and peaks for DP 2/3).

$$\text{triad strength} = \frac{I_{DP3} + I_{DP6}}{I_{DP2} + I_{DP4} + I_{DP5}} \qquad (1)$$

The profiles of the AaChio products detected by MS also differed between the three types of chitosan (Supplementary Fig. 1a). Although absolute quantification is not possible due to differences in ionization efficiency between the oligomers, the data allow us to search for differences between samples, revealing that higher proportions of trimeric products ($A_1D_2$ and $A_2D_1$) were released from the HTDA chitosan compared to the others, consistent with the RI signals. We used these data to calculate the number average A- and D-block sizes (Supplementary Fig. 1b). Whereas the A-block sizes were comparable for all samples, the D-blocks were smaller for the HTDA chitosan. To rule out the unlikely possibility that these differences originate from different starting DP values, reflecting different weight average molecular weights ($M_w$) of the substrates (Supplementary Table 1), we compared AaChio products of HTDA and CNA samples with the same FA (0.17) and a similar $M_w$ of ~110 kDa. The resulting RI chromatograms revealed the same trends in the FA = 0.17 samples (Supplementary Fig. 2) as seen for the FA = 0.32 samples (Fig. 1a). The corresponding triad strength values for the HTDA and CNA samples were 3.61 and 0.63, respectively.

Next, we checked whether these striking differences are indeed caused by the different production methods by analyzing a large set of HTDA, HMDA and CNA chitosans (Supplementary Table 1) spanning a wide range of FA values. Samples from different chitin sources were obtained from different producers, including chitosans that were deacetylated by freeze-pump out-thaw (FPT) cycles[36]. This confirmed that the previously observed differences are valid without exception: the triad strength was < 1 for all CNA chitosans over the whole FA range of 0.07−0.57 (Fig. 2) whether they were produced at the University of Münster or University of Lyon, or by the commercial supplier Heppe Medical Chitosan (HMC). The HMDA chitosans called Viscosans, produced by Flexichem AB, also showed low triad strength values. In contrast, HTDA samples from two different commercial suppliers showed triad strength values up to > 5. For sample series taken over the course of a HTDA reaction, it became apparent that the triad strength increased over the course of the reaction as the FA decreased, with the triad strengths of early samples being comparable to those of HMDA

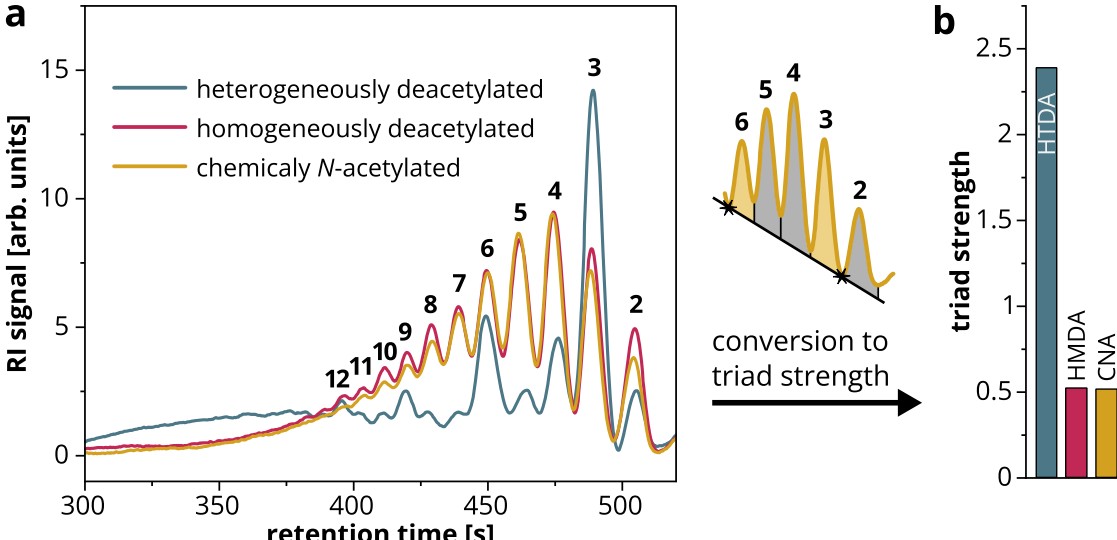

**Fig. 1 | Products of the AaChio-catalyzed hydrolysis of HTDA, HMDA and CNA chitosans (FA = 0.32).** Details of the chitosan samples are provided in Supplementary Table 1 (HTDA, shrimp_100min; HMDA, Viscosan_DDA69; CNA, 134_0.29). **a** Refractive index (RI) signals of the oligomer products after size exclusion chromatography (numbers above peaks indicate the DP). **b** Triad strength calculated from the RI signals based on Eq. (1). HTDA heterogeneously deacetylated, HMDA homogeneously deacetylated, CNA chemically *N*-acetylated.

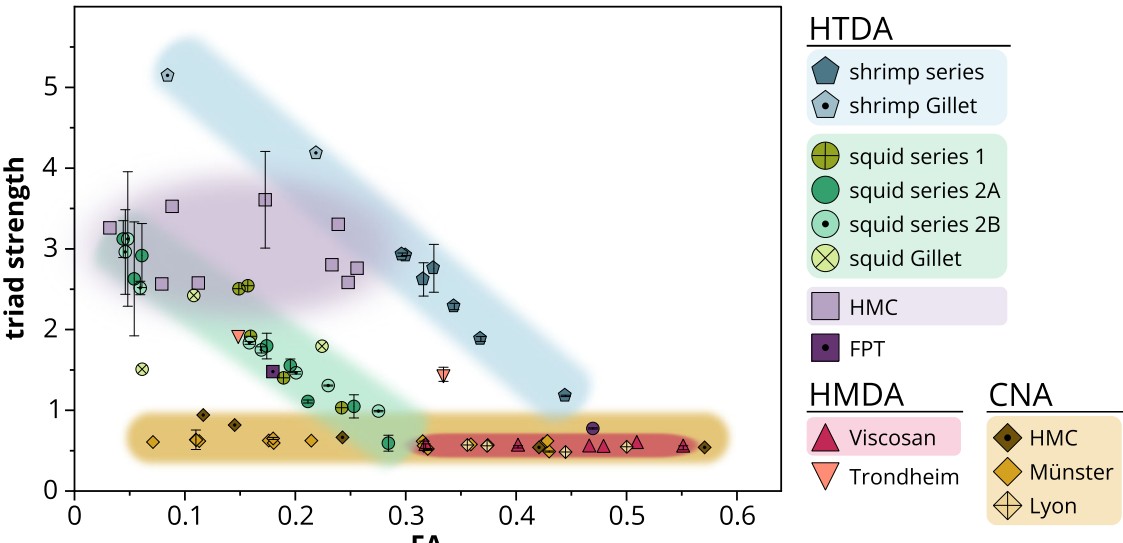

**Fig. 2 | Triad strengths of HTDA, HMDA and CNA chitosans following AaChio-catalyzed hydrolysis.** Details of the chitosan samples are provided in Supplementary Table 1. Samples labeled *series* were collected during heterogeneous deacetylation: series 1 and 2A were not subject to further chemical treatment, whereas series 2B samples are series 2A after chemical depolymerization. The triad strength value was calculated from the RI signals according to Eq. (1), for averages of multiple replicates, the standard deviation is indicated. The number of measurements ranges from 1–3 per sample, the exact number of replicates for each sample is listed in the source data of this figure (see data availability). HTDA heterogeneously deacetylated, HMDA homogeneously deacetylated, CNA chemically *N*-acetylated, HMC Heppe Medical Chitosan, FPT freeze-pump out-thaw.

and CNA chitosans. In addition, we noticed differences in the deacetylation series produced from shrimp (α-chitin) and squid (β-chitin) starting materials. As expected from previous studies[15], HTDA is more efficient for the less crystalline β-chitin (Supplementary Fig. 3), whereas the effect of high triad strengths at low FA values is more pronounced for chitosans derived from α-chitin. Samples prepared by the more efficient FPT process (HTDA)[36] of β-chitin behaved like other squid-derived HTDA chitosans. Interestingly, HMDA chitosans prepared from shrimp material by the former Kjell Vårum group (Trondheim) show a behavior intermediate between shrimp-derived HTDA chitosans and CNA chitosans or HMDA chitosans from Flexichem AB (Viscosans).

Again, the differences between the HTDA, HMDA and CNA chitosans were also visible in the altered AaChio product profiles, which in turn lead to deviating number average A- and D-block sizes (Supplementary Fig. 5). For HMDA and CNA chitosans, the A-block size increases and D-block size decreases with increasing FA in a near linear manner over the entire FA range. HTDA chitosans show comparable block sizes for intermediate FA values > 0.3, but the A-block size is slightly higher and the D-block size slightly lower compared to CNA chitosans at lower FA values.

Overall, these data show striking differences between HTDA and HMDA/CNA chitosans, even for samples with comparable average FA and DP values. The strongest indicator is an overrepresentation of

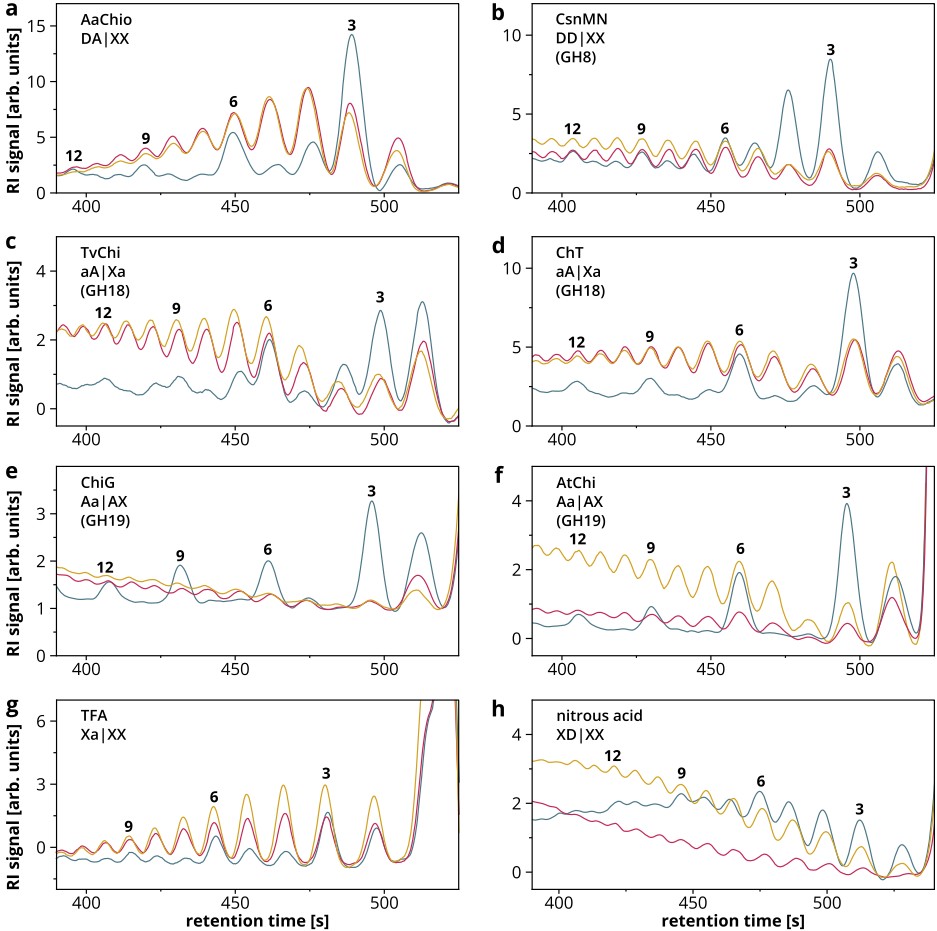

**Fig. 3 | Enzymatic and chemical cleavage products of HTDA (blue), HMDA (red) and CNA (yellow) chitosans (FA = 0.32).** Details of the chitosan samples are provided in Supplementary Table 1 (HTDA, shrimp_100min; HMDA, Viscosan_DDA69; CNA, 134_0.29). The refractive index (RI) signals of the oligomer products are shown after size exclusion chromatography. Enzymatic hydrolysis was carried out using **a** chitinosanase from *Alternaria alternata*[38] (AaChio), **b** chitosanase from *Bacillus* sp. MN[47–49] (CsnMN), **c** chitinase from *Trichoderma virens*[50] (TvChi), **d** human chitotriosidase[51–53] (ChT), **e** chitinase from *Streptomyces coelicolor* A3(2)[41] (ChiG), and **f** chitinase from *A. thaliana* (TAIR: AT3G54420) (AtChi). Partial chemical degradation was carried out by **g** acidic hydrolysis using trifluoroacetic acid (TFA),

or **h** nitrous acid deamination. The subsite preferences of the enzymes from subsites −2 to +2 are indicated with X (no strong preference), A (absolute specificity for A), a (considerable preference for A), D (absolute specificity for D) or d (considerable preference for D). Preferential chemical acidic hydrolysis after A-units[54,55] by TFA (corresponding to a considerable A-preference at subsite −1) as well as the cleavage by nitrous acid deamination[30,31] exclusively after D-units[56] (corresponding to an absolute D-specificity at subsite −1) are indicated accordingly. HTDA heterogeneously deacetylated, HMDA homogeneously deacetylated, CNA chemically *N*-acetylated.

DP = 3n products, quantified by an increase in the triad strength, which was only observed for HTDA chitosans, and was stronger at lower FA values and more pronounced for chitosans derived from α-chitin.

**Differences between chitosans are also apparent in chemical and other enzymatic cleavage products**

Next, we investigated the origin of the differences between HTDA and HMDA/CNA chitosans more closely by cleaving the various FA = 0.32 samples using enzymes with different subsite preferences, as well as trifluoroacetic acid (TFA) and nitrous acid. The products were analyzed by SEC-RI-MS and the corresponding chromatograms are shown in Fig. 3. As discussed for the AaChio hydrolysates (Fig. 3a), HTDA chitosans were again cleaved into products where DP = 3n was overrepresented, but the strength of the effect varied between reactions. It was weakest for chitosanase CsnMN from *Bacillus* sp. MN[47–49] (Fig. 3b), which strongly prefers D-units at subsites -2 and -1, but more pronounced for the GH18 chitinases TvChi from *Trichoderma virens*[50] (Fig. 3c) and human chitotriosidase (ChT)[51–53] (Fig. 3d), both of which have absolute specificity for A-units at subsite -1. The effect was particularly strong for products of GH19 chitinases such as ChiG from

*Streptomyces coelicolor* A3(2)[41] (Fig. 3e) and AtChi from *Arabidopsis thaliana* (TAIR: AT3G54420) (Fig. 3f), both of which have absolute specificity for A-units at subsites -2 and +1. Interestingly, the preferred formation of DP = 3n products was visible not only in the enzymatic hydrolysates but also following the partial chemical acid hydrolysis of HTDA chitosans using TFA (Fig. 3g). Given that A-A bonds and A-D bonds are hydrolyzed three times faster than D-A and D-D bonds under acidic conditions in concentrated HCl[54,55], TFA hydrolysis corresponds to the behavior of an enzyme with a high preference for A-units at subsite -1. In contrast, depolymerization by nitrous acid deamination, occurring exclusively after D-units[56], did not lead to pronounced peaks of DP = 3n products for the HTDA substrate (Fig. 3h). Instead, products of DP 7-9, 10-12, or 13-15 each occur with similar abundance, which is visible by similar heights of the RI peaks within each of these three DP groups, resulting in three steps in the RI chromatogram between 400–420 s, 420–440 s, and 440–470 s. In contrast, the DP distribution for products on HMDA or CNA chitosans was again uniform. Similar to the DP distributions, the product profiles of all six enzyme reactions and of TFA hydrolysates also differed between HTDA and HMDA/CNA chitosans whereas differences between the substrates were negligible

for nitrous acid deamination products (Supplementary Fig. 6). Furthermore, HTDA chitosan was hydrolyzed more efficiently by ChiG and nitrous acid, and less efficiently by TFA, AaChio, ChT and TvChi, compared to HMDA and CNA chitosans, whereas the cleavage efficiency of all three chitosans was comparable for CsnMN and AtChi (Supplementary Table 2). Finally, the same trends were visible when HTDA and CNA chitosans of the same FA (0.17) and DP ($M_w$ of ~110 kDa) were degraded (Supplementary Fig. 7).

In summary, the aberrant behavior of HTDA chitosans is visible in all cleavage products, but the strength of the effect depends on the type of cleavage. This confirms that triad formation is not caused by the enzyme but is a property of the chitosan sample. Nevertheless, the sample must be cleaved with certain specificity in terms of A-units and D-units to reveal the uniqueness of the HTDA samples, expressed by the irregular distribution of product DP values. Different average FA and/or DP values have been excluded as potential differences between the differently produced samples, and it would also be difficult to explain how these factors or their dispersity values ($Đ_{FA}$ and $Đ_{DP}$) could lead to a higher proportion of trimer and hexamer products. The only rational explanation is a PA in HTDA chitosans that differs from the random PA of HMDA or CNA chitosans and that possesses a certain regularity, which is reflected in the overrepresented products of the regular interval DP = 3n as well as in the similar amounts of products with DP 7–9, 10–12, or 13–15 obtained by depolymerization using nitrous acid.

## HTDA and HMDA/CNA chitosans and their hydrolysates show differences in bioactivity

To determine the significance of the more regular PA in HTDA chitosans in terms of bioactivity and potential applications, we tested the HTDA, HMDA and CNA chitosans (FA = 0.32) and their hydrolysates produced by chitosanase CsnMN[47–49] and chitinase TvChi[50] for the ability to elicit an immune response in *Arabidopsis thaliana* seedlings. Compared to hydrolysates of HMDA/CNA chitosans, CsnMN cleaved the HTDA sample less efficiently and released small products of DP 2–6 that are more strongly deacetylated. TvChi was also less active on HTDA chitosan, and the respective DP 2–6 products appeared more highly acetylated than in the respective HMDA/CNA hydrolysates (Supplementary Fig. 8).

Oxidative burst assays were used to measure the amount of reactive oxygen species produced directly after treatment of *A. thaliana* seedlings with different concentrations of the polymers or hydrolysates as a central marker for an induced disease resistance reaction (Fig. 4). The HTDA polymer triggered the strongest oxidative burst and caused elicitation at low concentrations whereas the CNA polymer and especially the HMDA polymer had a weaker effect and

higher concentrations were required for efficient elicitation (Fig. 4a). An endotoxin test (Supplementary Fig. 9) determined similarly low, negligible endotoxin concentrations[57] in all samples, ruling out the possibility that the differences in elicitation activity are caused by different endotoxin levels in the samples. Moreover, the stronger elicitation activity of HTDA polymers in comparison to CNA polymers was confirmed by further oxidative burst assays using potato (*Solanum tuberosum*) leaf discs (Supplementary Fig. 10). The CsnMN hydrolysate of the HTDA sample was similar in activity to the undigested polymer, but the CsnMN hydrolysis of the HMDA and CNA samples increased the oxidative burst (Fig. 4b). In contrast, enzymatic hydrolysis with TvChi reduced the elicitation activity of all three samples, especially the HTDA sample (Fig. 4c).

These results suggest that the more regular PA in HTDA polymers leads to a stronger immune response in plants compared to chitosan polymers with a random PA. The cleavage at deacetylated sites by CsnMN releases products with lightly acetylated ends and highly acetylated centers that trigger receptor responses[5,58]. In contrast, the destruction of these elicitation active acetylated patches by TvChi drastically reduces the oxidative burst[58]. Interestingly, hydrolysis of the HTDA sample using TvChi completely eliminated the initially very high bioactivity of this chitosan, although only limited hydrolysis occurred and a considerable amount of polymer remained. Possibly, the high elicitation activity just originates from certain, presumably highly acetylated patches of the HTDA polymers, and exactly these include motifs that are preferentially cleaved and therefore destroyed by TvChi. The overall differences in bioactivity indicate that the production of chitosans using different methods is interesting not only from an analytical perspective but also in the context of applications.

## In silico models suggest A-units are present at every third position in HTDA chitosans

Our data strongly suggest that HTDA chitosans, contrary to current opinion, have neither a random nor a block-wise PA, but rather one with certain regularity, especially at lower FA values. But what PA would explain the preponderance of trimers, hexamers and nonamers? Deducing this from the products of CsnMN, TvChi, ChT, ChiG and AtChi is difficult because these enzymes show complex cleavage behaviors due to combinations of absolute specificities and preferences for A-units or D-units at certain subsites. Therefore, we focused on the DP = 3n products of AaChio and chemical TFA hydrolysis. AaChio cleaves exclusively after the motif DA[38], whereas chemical acid hydrolysis preferentially occurs after an A-unit[54,55]. Accordingly, the abundant DP = 3n products most often have an A-unit at their reducing end, corresponding to the patterns XXA, XXXXXA and XXXXXXXXA. For AaChio, the A-unit is preceded by a D-unit (XDA,

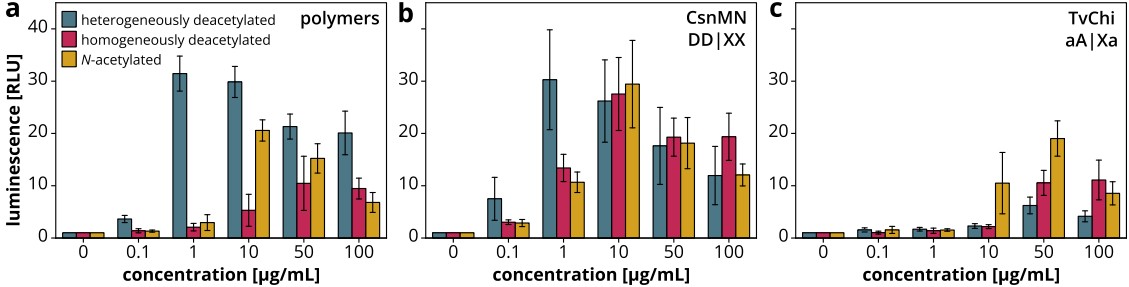

**Fig. 4 | Elicitation activity of FA = 0.32 polymers and corresponding hydrolysates on *A. thaliana* seedlings.** Details of the chitosan samples are provided in Supplementary Table 1 (HTDA, shrimp_100min; HMDA, Viscosan_DDA69; CNA, 134_0.29). Three differently produced **a** chitosan polymers as well as the corresponding enzymatic hydrolysates of **b** the chitosanase CsnMN[47–49] or **c** the chitinase TvChi[50] were tested at different concentrations. The subsite preferences of the enzymes from subsites −2 to +2 are indicated with X (no strong preference), A

(absolute specificity for A), a (considerable preference for A), D (absolute specificity for D) or d (considerable preference for D). For information on the composition of the hydrolysates see Supplementary Fig. 8. The oxidative burst was measured as a chemiluminescence signal resulting from the oxidation of luminol by reactive oxygen species. Shown are the averages of three independent experiments ($N$ = 3) of four replicates each ($n$ = 4) and their standard deviations.

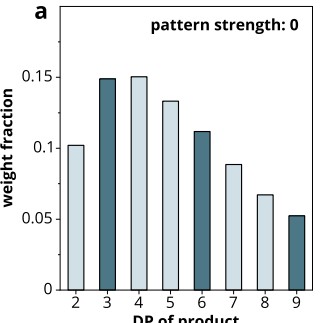
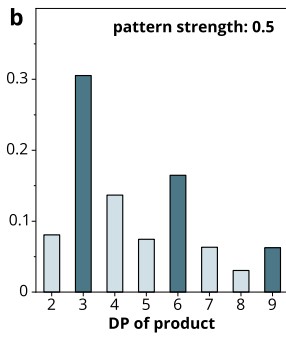
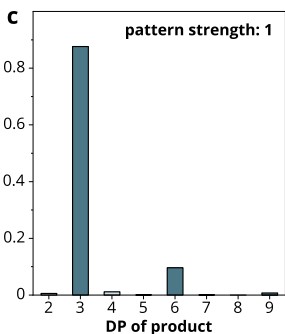

**Fig. 5 | Relative amounts of AaChio products with different DP values modeled in silico.** A simulated complete hydrolysis was performed with a cleavage specificity of DA | XX on 500 molecules each of DP 1000. The average FA of each substrate was 0.32, but the strength of the $(DDA)_{DP/3}$ pattern was varied. **a** The pattern strength of 0 corresponds to a completely random PA. **b** The pattern strength of 0.5 results in a moderate overrepresentation of A-units at every third position. **c** The pattern strength of 1 results in a maximum overrepresentation of A-units at every third position.

XXXXDA and XXXXXXXXDA). These products could in turn originate from a polymer in which A-units are overrepresented at every third position. We are not proposing that HTDA chitosans comprise the perfectly regular PA of $(DDA)_{DP/3}$ or even $(XXA)_{DP/3}$ but that the chitosans deviate from random and lean toward this more regular PA.

To model this, we used Python-based in silico methods to generate libraries of chitosan polymers with different average FA values and introduced different options for non-random PAs. The focus was on the overrepresentation of A-units at every third position to different extents, quantified by the parameter *pattern strength* ranging from 0 for a random PA to 1 for the PA closest to $(DDA)_{DP/3}$ for the corresponding FA. The cleavage of these chitosan libraries was then simulated at specific positions corresponding to the subsite preferences of different enzymes (e.g., after each DA-motif to simulate hydrolysis by AaChio)[38]. The molar amounts of products of a certain DP released by AaChio from FA = 0.32 chitosans with different pattern strengths were then modeled in silico. For chitosan with a random PA, the products were evenly distributed with respect to DP (Fig. 5a). However, as soon as the overrepresentation of A-units at every third position was introduced (pattern strength 0.5), products of DP = 3n became more abundant compared to the products of neighboring DP values (Fig. 5b). This effect became more prevalent with increasing pattern strength up to the maximum value of 1 (Fig. 5c).

We also modeled how samples from later in the HTDA reaction might behave by limiting the deacetylation of units at every third position (which we named the *pattern units*) to FA = 0.75 while the overall FA decreased from 0.5 to 0.3. The extent of the overrepresentation of DP = 3n AaChio products increased with the decreasing average FA of the substrate (Supplementary Fig. 11), as observed in vitro. Similar to the simulated cleavage with AaChio, the in silico models for other hydrolysis reactions indicated that a PA deviating from random toward $(DDA)_{DP/3}$ (pattern strength > 0) will lead to an overrepresentation of DP = 3n products (Supplementary Fig. 12). It should be noted that for all in silico reactions, simplified absolute specificities of the enzyme families and the acid were assumed at certain subsites, and possible preferences at other subsites were ignored.

If the PA of HTDA chitosans is indeed this regular, why do previous studies conclude that HTDA chitosans have either a random[14,26,27,30,35] or a block-wise PA[19,32–34], or combinations thereof[15,31,36]? First, there are authors that justify their conclusions based on the different physicochemical properties and enzymatic degradability of the chitosans they tested[15,19,32,33]. We argue that this deviating behavior may indeed indicate a non-random PA but not necessarily a block-wise PA as previously concluded. Second, DP distributions have previously been analyzed by SEC in the small degradation products of differently produced chitosans following nitrous acid deamination[30,31], which results in cleavage exclusively after D-units[56]. We modeled FA = 0.32 chitosan libraries

with a random PA (pattern strength 0) as well as those with pattern strengths of 0.5 and 1, cleaved them in silico after D-units, and compared the DP distributions of the products (Supplementary Fig. 13). Just as previous studies reported only small differences in the distributions of products of DP 1-6 for HTDA and HMDA chitosans and Bernoullian models for random PA[31], we found only small differences for pattern strengths of 0 and 0.5. Even the maximum pattern strength of 1 does not lead to the striking deviations which can be observed for in silico modeled small products of enzymatic or TFA hydrolyses (Supplementary Fig. 12). Whereas the earlier studies claim that this indicates a random PA, we conclude that the method is not able to detect the more regular pattern as suggested here. Our in vitro data confirms, that whereas cleavage by certain enzymes or TFA is suitable to reveal PA differences between the samples by looking at products of DP 2-6, this is not the case for nitrous acid deamination (Fig. 3). Nevertheless, it is indeed possible to unveil the more regular pattern in HTDA chitosans by nitrous acid deamination, but it is necessary to consider products which are larger than DP 6. The three steps observed in the RI signals of deamination products of HTDA chitosan for DP 7-9, 10-12, or 13-15 resulting from similar amounts of products within these three DP groups described above (Fig. 3) can be replicated in in silico models of the deamination products of chitosans with a moderate overrepresentation of A-units at every third position (Supplementary Fig. 14). Next to the overrepresentation of DP = 3n products, also this second type of non-uniform DP distribution in cleavage products of HTDA chitosans matches the in silico models for substrates with $(DDA)_{DP/3}$ pattern, which further supports our hypothesis of this more regular PA.

Third, ¹H-NMR and ¹³C-NMR analysis of HTDA chitosans yielded diad frequencies comparable to those calculated for a random PA[14,26], or PA values in the random-dominated range (0.5 < PA value < 1.5)[27]. To investigate this, we modeled chitosan populations with different pattern strengths over the entire FA range from 0 to 1, and used them to calculate the PA values as previously described[27,28] (Supplementary Fig. 15). Interestingly, the maximum PA value (pattern strength 1, FA = 0.33), corresponding to the ideal $(DDA)_{DP/3}$ pattern, is only 1.5, so still within the previously defined random-dominated range[27]. Furthermore, most commercial HTDA chitosans have FA values of ≤ 0.2 because customers favor low acetylated products. In combination with moderate pattern strengths, which we suspect to be more likely, the calculated PA values deviate only negligibly from 1 (random PA). Therefore, diad analysis is appropriate to distinguish between alternating, random and block-wise PA[27,28], but cannot reveal longer repeated motifs, such as the pattern proposed here. Indeed, others have already suggested that diad analysis may report a random PA for certain block structures[26,37], which is why the NMR analysis of triad frequencies is also required (not to be confused with the terms triad

strength and triad formation in our study). Again, earlier authors predicted random PAs for HTDA chitosans but they also reported that the method typically has errors of at least 15%[26]. Our in silico models showed only slight deviations from random triad frequencies for small or intermediate pattern strengths (Supplementary Table 3), which fall within the typical error margins of the method. Accordingly, we conclude that both NMR-based diad and triad analysis are unable to detect the more regular PA in HTDA chitosans proposed herein. Indeed, the higher sensitivity of PA analysis based on MS fingerprinting compared to NMR-based PA analysis was already reported and discussed previously[37].

## Discussion

We observed striking differences between the HTDA and HMDA/CNA chitosans at the structural and functional levels. Hydrolysates of all HTDA chitosans were enriched in DP = 3n products, although the extent of overrepresentation varied for different FA values and enzymes. In contrast, the hydrolysates of CNA and HMDA chitosans consistently showed a continuous distribution of DP values. Combining the in vitro experimental and in silico modeling data, we propose that HTDA chitosans exclusively feature a non-random but non-block-wise PA that deviates from random and tends toward the pattern (DDA)$_{DP/3}$, where A-units are overrepresented on every third unit. Our hypothesis is supported by earlier findings[59] revealing over-represented DP = 3n products following the enzymatic digestion of HTDA chitosan, including the patterns (DXY)$_1$DAA, (DXY)$_2$DAA and (DXY)$_3$DAA (where X = A and Y = D or vice versa) for DP 6, 9 and 12, respectively. Our proposed (DDA)$_{DP/3}$ pattern, in combination with the enzyme's subsite preference, would in fact lead to products such as (DDA)$_n$DAA or (DAD)$_n$DAA. In addition, another group observed overrepresented DP = 3n products in digestions of HTDA chitosans with acidic mammalian chitinase (GH18)[60,61], whereas HMDA chitosans yielded uniform DP distributions after enzymatic cleavage[61].

The origin of this special, regularly repeating PA in HTDA chitosans is unclear, although the degree of crystallinity in the original chitin material seems to play a role because the effect is more pronounced in chitosans derived from the more crystalline α-chitin rather than β-chitin. Moreover, homogeneous conditions during the production of HMDA chitosans by the former group of Kjell Vårum (Trondheim)[39,62] may lower the crystallinity due to swelling, resulting

in chitosans with a PA closer to random. Accordingly, HMDA samples from Flexichem AB (Viscosans) prepared using an optimized homogeneous process feature a fully random PA. If the regular PA is indeed formed during the HTDA reaction, there must be a molecular-level mechanism protecting every third position from efficient deacetylation. Decades ago, studies proposed that HTDA efficiently deacetylates the amorphous regions of chitin, while only slightly affecting the more crystalline portions[15,16,19]. This reaction can be described by the shrinking core model[63,64], in which deacetylation occurs at the defined interface of the liquid alkali reagent and the solid chitin particle, leading to a shrinking core of still acetylated chitin that has yet to react. Simultaneously, the deacetylated chitosan forms a growing layer around the core and increasingly hinders inward hydroxide ion diffusion. The model explains why HTDA chitosans are suspected to have high FA dispersity ($Đ_{FA}$) values[65] and why the efficiency of deacetylation levels off over the course of the reaction[15], as we also observed (Supplementary Fig. 3). However, the shrinking core model does not explain how a regular PA could be formed. An alternative hypothesis for the cessation of deacetylation is the formation of a chitosan amide anion as part of the reaction mechanism[66] (R-N⁻, Fig. 6a). The study suggests that the anions' accumulated negative charge electrostatically repels hydroxide ions that are necessary for subsequent deacetylation of adjacent residues (Fig. 6b). Moreover, the anion is stabilized by the partially cationic acetamide group carbon of a neighboring chitosan chain which in turn reduces the electrophilicity and hence the probability of deacetylation of the latter (Fig. 6c). This implies the formation of a regular secondary structure resulting from the intermolecular interactions of chitosan molecules during the HTDA reaction which, in turn, could account for the protection of specific positions from deacetylation and the formation of a regular PA (Fig. 6d). Another possible explanation is the formation of a stable interaction between a new D-unit and sodium acetate immediately after deacetylation, thus inhibiting the efficient deacetylation of surrounding units[67], potentially leading to similar secondary structure and protection effects. In both cases, hydration (or more precisely the hydration status of hydroxide ions or chitin) may govern the reactivity. In conclusion, the HTDA reaction is complex and any or all of the reasons proposed above may explain the assumed high FA dispersity ($Đ_{FA}$), the cessation of the deacetylation reaction, and the formation of a more regular PA.

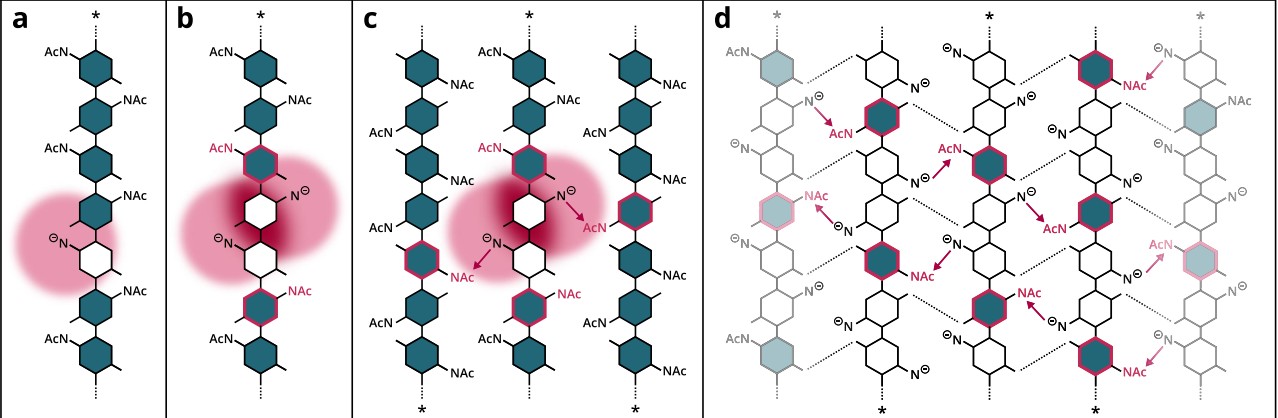

**Fig. 6 | Hypothetical molecular mechanisms for the formation of a more regular PA.** A-units are shown in blue, D-units in white and the reducing end side of the chain is highlighted (*). **a** A single deacetylation leads to the formation of an amide anion (R-N⁻, negative charge highlighted in pink). **b** Two deacetylated amide anions lead to a strong negative charge density amidst the chain which electrostatically repels hydroxide ions that may attack adjacent A-units, protecting the latter from deacetylation (pink stroke of A-units). **c** Amide anions are stabilized by N-acetyl groups of neighboring chains, hence leading to the protection of these groups from deacetylation. **d** A complex network of interactions may be formed between adjacent chains, including interactions that protect certain N-acetyl groups from deacetylation.

The hypothesis of a chitosan secondary structure during the HTDA reaction is supported by in silico models of the triad strength parameter for chitosans of different FA and different strengths of the $DDA_{DP/3}$ pattern (Supplementary Fig. 16). Just as the PA value is most sensitive to detect an alternating PA at FA 0.5[37], the triad strength parameter is most sensitive to unveil the overrepresentation of A-units at every third position between FA 0.3-0.4. Especially for strong pattern strengths, it becomes obvious that very high triad strength values are only possible for chitosans of FA < 0.45, but it is still feasible to differentiate between weak and strong pattern strengths in high FA chitosans by this parameter. When comparing the measured triad strengths for the HTDA samples derived from shrimp or squid chitin (Fig. 2, highlighted in Supplementary Fig. 16) with the in silico trends, we can conclude that not only the FA decreases and the triad strength increases during the HTDA reaction, but also the pattern strength increases. Samples taken early in the reaction show weak or no patterns, indicating that the $DDA_{DP/3}$ pattern is not formed during initial chitin deacetylation but only when already a considerable number of units is deacetylated. In accordance with the hypothetical molecular mechanism elaborated above, the pattern formation starts once chitosans become soluble and are able to form secondary structures different from the crystalline chitin arrangement. Possibly, the pattern formation starts later in the HTDA reaction for squid-derived chitosans, because due to the lower crystallinity of β-chitin, the deacetylation is already further progressed before secondary structures have been formed.

We note that the deviating PA in HTDA chitosans indicates neither inferior nor superior quality compared to HMDA and CNA chitosans. Rather, it is necessary to acknowledge these differences in PA, and that they lead to differences in chitosan bioactivity. Our assays with *A. thaliana* seedlings showed that the regular-PA HTDA polymer is a stronger elicitor than the random-PA HMDA and CNA chitosans of the same FA, suggesting that chitosans with a PA favoring $(DDA)_{DP/3}$ might be recognized particularly well by plant receptors[5]. Furthermore, all enzymatic hydrolysates of HTDA chitosans exhibited non-uniform DP distributions and product profiles deviating from those of HMDA and CNA chitosans. This is important because chitosans are usually processed by enzymes such as human ChT (in biomedical applications) or chitinases present in the soil microbiome or in plants (in agricultural applications). Our data suggest that even chitosans with the same average DP and FA and similar dispersities thereof can vary significantly in performance due to the different PAs resulting from different production methods. Thus, the special, regular PA we identified may be advantageous in some applications but a disadvantage in others.

Further research is needed to verify (or reject) the hypothesis of the $(DDA)_{DP/3}$ pattern formation and to find its molecular basis during the HTDA reaction. Analyzing more deacetylated HMDA or HTDA chitosans by AaChio hydrolysis and subsequent SEC-RI-MS could be useful, such as chitosans prepared using different process parameters, from different reaction time points, or from different chitinous starting materials. More importantly, it is necessary to examine the potential emergence of secondary structures during the HTDA reaction, which we suspect are responsible for the protection of every third unit from deacetylation.

In contrast to current opinion, we conclude that HTDA chitosans, including the majority of commercially available chitosans, have a non-random, non-block-wise PA and therefore differ considerably from random-PA chitosans produced by other methods. This is important because the PA influences the enzymatic processing, physicochemical properties[1,37,68], and bioactivities[69] of chitosans, thus affecting the performance of commercial products as well as the chitosans used for research purposes. Therefore, in addition to the structural parameters DP and FA (and ideally the corresponding dispersities, $Đ_{DP}$ and $Đ_{FA}$), the process used to prepare chitosans (HTDA, HMDA or CNA) should be taken into account when investigating chitosan properties and bioactivities, and mentioned when reporting the obtained results.

## Methods

### Chitosan samples

The chitosan samples used in this study were either produced in-house or provided by Dr. Katja Richter (Heppe Medical Chitosan, Germany), Dr. Mats Andersson (Flexichem AB, Sweden), Gillet Chitosan SAS (France), or in 2003 by the former group of Prof. Kjell Vårum (Norwegian University of Science and Technology, Trondheim). Supplementary Table 1 provides detailed information about the origin, characteristics, production method, and chitin source. The FA was determined by enzymatic MS fingerprinting[25] (adapted method[70] for insoluble chitin samples), and the weight average and number average molecular weight ($M_w$ and $M_n$, respectively) and dispersity ($Đ_M$) were analyzed by SEC-MALLS-RI[20]. The in-house HTDA samples were deacetylated at 90 °C using 50% (w/v) NaOH for the indicated amount of time. Selected samples (series 2A) were then chemically partially depolymerized to obtain series 2B. FPT HTDA samples were prepared in-house[36], as were the CNA samples produced in Lyon and Münster[20,21]. The chitosans to be hydrolyzed with AaChio were dissolved overnight in ammonium acetate (200 mM, pH 4.2; Roth) whereas for other reactions, the chitosans were dissolved overnight in defined amounts of acetic acid (Roth) corresponding to a molar ratio of acid to free chitosan amino groups of 1.2. Subsequently, dissolved chitosans were filtered with nylon centrifugal filters (pore size: 0.2 μm; VWR) to remove any insoluble particles.

### Production of chitosan cleavage products

The chitosans were hydrolyzed with AaChio[38] until the end point[37]. Selected chitosans (Supplementary Table 1) with FA = 0.32 (HTDA, shrimp_100min; HMDA, Viscosan_DDA69; CNA, 134_0.29) or FA = 0.17 and $M_w$ of ~110 kDa (HTDA, 80/20; CNA, 134_0.19) were also hydrolyzed at a concentration of 1 g/L with five other enzymes at 37 °C: CsnMN[47–49], TvChi[50], ChT[51–53], ChiG[41], and AtChi (TAIR: AT3G54420). Enzyme expression and purification as well as the precise reaction conditions for hydrolysis are provided in Supplementary Table 4. Partial chemical hydrolysis under acidic conditions followed the protocol for monosaccharide analysis[71]. We mixed 20 μL of dissolved chitosans (1 g/L) with 1 mL TFA (2 M; Roth), autoclaved the reaction (45 min, 121 °C) and air-dried the products at 40 °C to remove the solvent. Twice, we added 500 μL of methanol (Sigma-Aldrich), followed by air-drying at 40 °C. The dried sample was dissolved in 30 μL of water for analysis. Partial cleavage by nitrous acid deamination was performed two times[72], with a GlcN unit/$NaNO_2$ molar ratio of 4 each. Briefly, 5 μL of freshly dissolved $NaNO_2$ (2.142 g/L) was added to 100 μL of acid-dissolved chitosans (1 g/L) and incubated at room temperature for 16 h under constant stirring. The $NaNO_2$ addition and incubation was repeated a second time.

### SEC-RI-MS analysis

All cleavage products were analyzed by SEC-RI-MS as described recently[43]. Briefly, 3 μg of each sample was injected and separated using an ACQUITY UPLC Protein BEH SEC column with a pore size of 125 Å (Waters Corporation), followed by RI detection using an ERC RefractoMax 520 (Thermo Fisher Scientific) and MS analysis using an amaZon speed ESI-$MS^n$ (Bruker). The MS data were analyzed using Data Analysis v4.1 (Bruker) and an in-house Python script based on the module pymzML[73]. Data Analysis v4.1 was also used to export RI data for further analysis using OriginPro 2023 (OriginLab). The calculation of block sizes was based on the AaChio products[37] (Supplementary Fig. 4, Supplementary Eq. (1) and Eq. (2)).

## Elicitation assays

For the elicitation assay in *A. thaliana*, three differently produced FA = 0.32 chitosans were hydrolyzed by adding 0.01 μg CsnMN or 0.1 μg TvChi per mg chitosan directly to the acidic aqueous chitosan solution and incubating at 37 °C for 3 days or overnight, respectively. The hydrolysis was stopped by heating at 95 °C for 5 min. The endotoxin concentrations were determined using a Pierce™ Chromogenic Endotoxin Quant Kit (Thermo Fisher Scientific) for all samples at a chitosan concentration of 200 μg/mL. The elicitation activity of the intact polymers and enzymatic hydrolysates was compared on *A. thaliana* seedlings[5]. Briefly, the samples were added to prepared seedlings and the response was measured by immediately quantifying the amount of produced $H_2O_2$ using luminol. Data represent the maxima of the burst curves. Similarly, elicitation activity was tested in potato (*Solanum tuberosum*) leaf discs[74].

## In silico modeling

Python scripts were developed to simulate the behavior of different substrates and their hydrolysates in vitro. The foundation was the modeling of populations of chitosan molecules with varying characteristics (DP, average FA, and PA, including the strength of the $(DDA)_{DP/3}$ pattern). The modeled populations were used to count triad and diad frequencies and to calculate the PA value from the latter, allowing for the simulation of diad-based PA analysis[14,26,27]. They were also used for in silico cleavage by various enzymes with different specificities, nitrous acid (cleavage after D-units), or TFA (cleavage after A-units). The Python scripts are available as a user-friendly free web tool at https://lcp-simulator.anvil.app[75].

## Reporting summary

Further information on research design is available in the Nature Portfolio Reporting Summary linked to this article.

## Data availability

Scatter plot, line plot and bar chart source data of Figs. 1–5 are provided with this publication and accessible at https://doi.org/10.17879/86998570305. Datasets presented in the supplementary information are accessible at https://doi.org/10.17879/46918475258. All other data are available from the corresponding author upon request.

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

## Acknowledgements

M.J.H. would like to thank the German Academic Scholarship Foundation, which provided financial support for a doctoral scholarship. The authors are grateful to Dr. Katja Richter (Heppe Medical Chitosan, Germany), Dr. Mats Andersson (Flexichem AB, Sweden) and Prof. Kjell Vårum (Norwegian University of Science and Technology, Trondheim) for the generous gift of chitosans. Special thanks to Prof. Martin Peter and Prof. Laurent David (University of Lyon, France) for very helpful critical discussions, and to Katharina Eickelpasch for providing oxidative burst data for potato leaf discs. We acknowledge Dr. Richard M. Twyman (Twyman Research Management) for editing the manuscript.

## Author contributions

D.G. and S.T. prepared samples, S.R. performed the elicitation experiments on *A. thaliana* and analyzed the results. M.J.H. performed all other experiments, developed and executed the Python scripts for in silico modeling, analyzed the data, and wrote the main manuscript. B.M.M. and S.C.-L. supervised the work. All authors discussed the results, developed hypotheses, and reviewed and revised the manuscript.

## Funding

## Competing interests

The authors declare no competing interests.
