## [Peer Review File · Nature Communications]

Heterogeneously deacetylated chitosans possess an unexpected regular pattern favoring acetylation at every third positionReviewers' Comments:

Reviewer #1:

Remarks to the Author:

This work provides insight into the chemical structure of partially acetylated chitosans obtained by heterogeneous deacetylation. This topic has been discussed for decades, as chitosans are widely regarded as biopolymers with random or rather blockwise distribution depending on the chemical process to synthesize them. Chemical composition in terms of pattern has so far been determined mainly by NMR spectroscopy when calculating the abundance of sugar diads and triads. Of note, the authors describe in this manuscript that the pattern is more regular, with a prevalence of (DDA)DP/3, and this is demonstrated by mass spectrometry fingerprinting experiments and in silico modelling. Besides the fraction of acetylated units, i.e. FA, and the molecular weight, which can be expressed as degree of polymerization, DP, the pattern of the two building sugars (glucosamine and N-acetylglucosamine) has been shown to play a pivotal role, especially with regard to bioactivity. In addition, heterogeneous deacetylation of chitosans is widely used commercially to produce partially acetylated chitosans from chitins of different origins. Therefore, besides the scientific message, the results reported in this manuscript have a certain potential for industry. In view of all these considerations, the manuscript could have scientific relevance.

Overall, the manuscript is well written and, as anticipated above, addresses a very important question in the field, even though there is no mechanistic answer to the molecular basis of the process that leads to this regular pattern. One simple question remains unanswered: how would heterogeneous deacetylation promote a regular pattern (DDA)DP/3? This is the major shortcoming of this manuscript. Although some speculation is made in the discussion of the results, the Reviewer believes that the molecular basis for this unexpected regular recurrence of PA in heterogeneously deacetylated chitosans needs to be demonstrated in this manuscript to make a fundamental breakthrough in this field.

Additional points are listed below:

1. Enzymatic digestion using a chitinase followed by mass spectrometry is a worth alternative to NMR for investigating the chemical compositions of chitosans. This was previously demonstrated by the authors (reference 36). It would be useful for the readership to provide additional information about the sensitivity of the technique with respect to standard NMR.
2. Line 94. It is not clear why the authors started this investigation using partially acetylated chitosans with FA = 0.32 deriving from heterogeneous/homogeneous deacetylation or re-N-acetylation.
3. Line 104: The authors introduce the parameter "triad strength" to obtain information about DP = 3 products after digestion. Why was the peak at DP = 9 excluded in the calculation of this parameter? While DP = 12 is almost the same for all investigated conditions and therefore would not play a significant role in the calculation, this would be the case for DP = 9, as it is significantly lower for the heterogeneously deacetylated situation (Figure 1a).
4. The authors must provide limitations of the triad strength parameter. For instance, taking a look at figure 2, it emerges that the parameter fails to be valid for chitosans showing a fraction of acetylated units in the range 0.4 - 0.6. This is particularly important, since chitosans showing this FAs are well-known to start to become soluble at neutral pH. Thus, in principle, a direct correlation between pattern (determined by triad strength/composition) and solubility can not provided for this set of chitosans. Thus, FA still would remain the main parameter to predict the solubility of chitosans at neutral pH.
5. Nitrous acid depolymerization should also be considered as control to validate the experiments reported in Figure 3, as it randomly cleaves the chitosan backbone in correspondence with the D-units, generating reactive 2,5-anhydro-D-mannose residues that can subsequently be reduced.
6. Figure 3f. It is not clear why trifluoroacetic-assisted depolymerization would generate less products with DP = 3n in the case of heterogeneous deacetylation. This is in contrast with enzyme-assisted depolymerization.
7. Bioactivity experiments. The authors claim that chitosans (and their products) from different chemical procedure elicit a different oxidative burst using *A. thaliana* as a model. This is interesting.

The authors must exclude that differences in terms of oxidative stress would be related to endotoxins present in chitosan samples rather than their chemical composition. LAL tests on chitosan samples could be useful to test this.

8. Experiments on bioactivity. It should be very interesting to carry out further experiments on bioactivity and compare chitosans from different chemical processes. What about oxidative stress in immune cells or antibacterial properties, for example?

Reviewer #2:

Remarks to the Author:

The manuscript is a very strong contribution to the important topics of chitosan synthesis, analysis, and bioactivity. It is also a strong contribution to the very knotty problem of determining and controlling sequence of monosaccharides in polysaccharides. Polysaccharides are one of a few great families of natural polymers, alongside proteins and poly(nucleic acid). The tools for understanding their structures are much less powerful than those in the hands of protein and DNA/RNA investigators, due in large part to the far, far greater complexity of polysaccharide structure. The manuscript by Moerschbacher et al. is an important contribution with potential impact across a number of scientific fields (e.g. chemistry, biology, agriculture, polymer science).

It is very well written, methodology is inventive and appropriate, and it is logically presented. I really only have two suggestions for improvement:

1) The statement on page 2 about chitosans being rare in nature is not really accurate. It certainly is true that there are not nearly as many cationic polysaccharides as there are neutral and anionic ones. However, chitosan is a common element of fungal cell walls, and fungi are quite abundant. Thus this statement should be revised or removed.

2) The structural arguments made about chitosan sequence following hydrolysis are complex, and I suspect will be difficult for some non-experts to follow. Especially since this is a journal with a broad scientific readership, it would be best if the arguments are presented with as high clarity as possible. I would suggest some use of structural cartoons, and of simple depictions of some of the proposed mechanisms by which regular patterns of acetylation are achieved, according to the hypotheses of the authors.

Very strong, novel, impactful piece of work.

Reviewer #3:

Remarks to the Author:

This is a nice paper that reaches an important conclusion that will be of interest to researchers working on the production and application of chitosan. In recent years, the Moerschbacher team has made major steps that are needed to "finally" be able to properly characterize chitosans and, thus, to connect chitosan properties to biological function. This is crucial in a field that is big but that, for long, has suffered from mediocre publishing. The prime observation, non-random acetylation of heterogeneously deacetylated chitosans, is important, and well documented by studies with multiple enzymes, multiple chitosans and modelling. The fact that the authors present data for different chitosans and enzymes is laudable and adds reliability and impact. The paper is well written and easy to follow, despite the rather specialized nature of the topic. I have a few concerns and some suggestions for improvement, and I point at some issues that require clarification. Some of my suggestions are meant to make this paper more accessible to readers who are less familiar with the field.

1. Line 17: "and found that both assumptions are wrong." This is confusing (because the reader may associate this with assumptions regarding chitosans produced by homogeneous deacetylation or chemical N-acetylation, which is not what the authors intend to say); please rephrase.

2. Line 25 & 38 & 88-90: In my view, one should not give the impression that chitosan occurs in nature and I would advise against using the term "enzymatically produced natural chitosans" in Line 25. Yes, partially deacetylated chitin occurs in nature, but chitosans, as I see these, so acid-soluble polymeric materials in which all chains (not only the surface of a fibril) are deacetylated, and that give viscous solutions, do not really occur in nature. It is not possible to dissolve chitin by treating with deacetylases.

Line 38 "Chitosans are rare in nature". Do they or do they not occur? And if yes, how then do the authors define "chitosan"? A high quality, reliable reference is needed here.

Line 88 & 90: "Our data also suggest that commercial chitosans fundamentally differ in their structure from natural chitosans produced by the action of chitin deacetylases, which are highly unlikely to generate the same, regular PA." This text must be changed for multiple reasons: (1) do natural chitosans even exist? (2) Deacetylases can deacetylate chitin but cannot convert chitin to chitosan, & (3) if they could there is no basis whatsoever to state that they are "highly unlikely to generate the same, regular PA" (the contrary is true; enzymes can do exactly this, in principle).

Please make related corrections throughout the manuscript.

3. Line 42-45: The use of the terms heterogeneous and homogeneous is jargon. By just reading the text as is, the two terms are confusing, for example considering the meaning of the term "homogeneous catalysis". I invite the authors to use a few more words to explain what is meant by these two terms and, importantly, to explain how exactly the two processes differ (they both seem "heterogeneous" to me). This is of particular importance since one key point of this paper is that the process used has impact on the pattern of deacetylation. (Nb. See also remark at Lines 141-143/332-334, below).

4. Line 72-74: Instead of, or next to, reference 38, the authors should cite some original work here, for example the pioneering work by Kjell Varum, who studied both GH18s and GH19s, some 15 years ago (e.g. Sorbotten 2005, Sasaki 2006, Horn 2006, Heggset 2009).

5. Line 74-77: This was not easy to follow for this reviewer. I suggest some rephrasing and perhaps some more words to clarify the issues for the less informed reader. Two specific comments: (a) Please add a statement about the (lack of) specificity of AaChio in subsites +1 and +2. (b) I do not understand this sentence: "This enzyme therefore hydrolyzes chitosans precisely next to each transition from D- to A blocks". What does this mean and how can this be inferred? What kind of "blocks" are we referring to here? And what kind of "transition"? This enzyme cuts after DA diads which, to me, does not seem linked to a "transition from D- to A blocks". I recommend rephrasing and some expansion of lines 74-77.

6. Lines 141-143 & 332- 334:

Here the authors point at HMDA chitosans prepared using a "less homogeneous process" that show intermediate behavior. I struggle with this for two reasons:

a. What does "less homogeneous" mean? Reference? Where can I find the recipe for this "less homogeneous" process? What exactly is the difference?

b. These Trondheim chitosans have been used to unravel processivity in chitinases (Varum, Eijsink et al., 2005 - 2010) and one of the key elements of data interpretation in those elegant studies was the assumption that these chitosans have random acetylation. Indeed, in those studies, there are no indications that the acetylation pattern is non-random (e.g., "intermediate"). Can these previous and the present observations be reconciled? Perhaps this should be discussed to some extent. Of note, these are just two chitosans of very many, so even deleting these would not really affect the paper or its conclusions.

7. Lines 232-239: I appreciate the differences that are demonstrated, but I find the discussion a bit too firm. Yes, there are differences and they should be pointed out, but I think that text like "For the latter, the cleavage of D-rich regions by CsnMN releases eliciting active products with A-rich centers, whereas the destruction of immunomodulatory A-rich parts by TvChi drastically reduces the elicitation activity*." goes too far. This is both unprecise and speculative. You cannot really connect the enzymes to "D-rich" and "A-rich" regions. What is an "A-rich center"? What is meant with "immunomodulatory A-rich parts"? Where is the evidence that these "parts" are "immunomodulatory"? What does immunomodulatory mean in this context? What does the * mean? Another concern is that TcChi reduces bioactivity a lot, while, as the author state "only limited hydrolysis occurred and a considerable amount of polymer remained." How can this be explained?

8. Table S2: More details need to be provided: were the reactions complete? How did the authors ensure this? Add information about the cleavage specificity of the enzymes. Add data for AaChio.

9. Legend to Fig. S5. "The faster chemical hydrolysis after A-units₄₈ is indicated accordingly." I have no idea what this means. You are not measuring rates here. Reference 48 does not belong here.

Other issues:

1. In the main text and in the supplementary material, there are several Figures and Tables showing data for chitosan degradation by different enzymes (Figs. 3, S5, S6; Table S2). I would have preferred if results for the AaChio were included in these Figures and Tables for reference. I realize that these results are shown elsewhere, but it is not so easy to compare.

2. In the manuscript, references are made to Kjell Varum and the group of Kjell Varum. Unfortunately, Kjell Varum died in 2017 and there is no longer such a thing as the group of Kjell Varum. I think that this somehow needs to be made clear in the paper. Please consider some rephrasing.

3. PA – please decide if the A is in subscript or not and check the complete manuscript (e.g., line 61).

4. Line 219: rephrase. Which tendency?

5. Fig. 4b,c: please show cleavage specificities on the two enzymes, similar to what is done in Fig. 3.

6. Line 248: A reference must be provided for the TFA cleavage specificity.

Point-by-point response to reviewers

Reviewer 1

This work provides insight into the chemical structure of partially acetylated chitosans obtained by heterogeneous deacetylation. This topic has been discussed for decades, as chitosans are widely regarded as biopolymers with random or rather blockwise distribution depending on the chemical process to synthesize them. Chemical composition in terms of pattern has so far been determined mainly by NMR spectroscopy when calculating the abundance of sugar diads and triads. Of note, the authors describe in this manuscript that the pattern is more regular, with a prevalence of (DDA)DP/3, and this is demonstrated by mass spectrometry fingerprinting experiments and in silico modelling. Besides the fraction of acetylated units, i.e. FA, and the molecular weight, which can be expressed as degree of polymerization, DP, the pattern of the two building sugars (glucosamine and N-acetyl-glucosamine) has been shown to play a pivotal role, especially with regard to bioactivity. In addition, heterogeneous deacetylation of chitosans is widely used commercially to produce partially acetylated chitosans from chitins of different origins. Therefore, besides the scientific message, the results reported in this manuscript have a certain potential for industry. In view of all these considerations, the manuscript could have scientific relevance.

Overall, the manuscript is well written and, as anticipated above, addresses a very important question in the field, even though there is no mechanistic answer to the molecular basis of the process that leads to this regular pattern. One simple question remains unanswered: how would heterogeneous deacetylation promote a regular pattern (DDA)DP/3? This is the major shortcoming of this manuscript. Although some speculation is made in the discussion of the results, the Reviewer believes that the molecular basis for this unexpected regular recurrence of PA in heterogeneously deacetylated chitosans needs to be demonstrated in this manuscript to make a fundamental breakthrough in this field. Additional points are listed below.

We would like to thank the reviewer for the positive feedback and the helpful remarks. Of course, trying to explain the molecular basis for the unexpected regular pattern will be the logical next step. We have discussed this point thoroughly with several experts in chitin and chitosan chemistry and crystallography (as mentioned in the Acknowledgements). We suggested a possible model (which we eventually also briefly described in the Discussion) which the experts found plausible, but they also made it clear that experimental proof would be a major task in itself and can be expected to take years (they reminded us that the development of the currently established modified shrinking core model of chitin deacetylation published in 2021¹ is based on a variety of studies on chitin deacetylation²⁻⁹ (not all cited here), the earliest dating back to the 1970s^{10-12!}). Also, this would most likely require the collaboration of experts from various backgrounds with extensive knowledge in chemical reaction mechanisms, crystalline structures, deacetylation kinetics, etc. Encouraged by the remarks of the reviewer, we have now extended the description of our hypothetical scenario of how the acetylation pattern forms during heterogeneous deacetylation, also including an extra figure to visualize the process in the Discussion. We are confident that this will serve as a suitable starting point for experimental verification in follow-up studies.

All our comments to the reviewer's point-by-point remarks are listed below and additionally, amends to the manuscript are indicated in the Word file of the revised manuscript using tracked changes.

- 1. Enzymatic digestion using a chitinase followed by mass spectrometry is a worth alternative to NMR for investigating the chemical compositions of chitosans. This was previously demonstrated by the authors (reference 36). It would be useful for the readership to provide additional information about the sensitivity of the technique with respect to standard NMR.**

Making accurate quantitative statements about the sensitivity of both the NMR-based and the MS-fingerprinting-based analysis of FA and PA of chitosans is currently not feasible given the lack of fully defined standards.

Still, our study shows that MS-fingerprinting is definitely more sensitive than NMR when it comes to detecting the $DDA_{DP/3}$ pattern, as shown in Supplementary Figure S15 and Supplementary Table S3 and elaborated in the corresponding part of the manuscript, especially in comparison with the new Supplementary Figure S16. Moreover, MS-fingerprinting has been reported to detect PA differences in enzymatically deacetylated chitosans with higher sensitivity compared to NMR. Therefore, and in accordance with the reviewers recommendation, we added the following sentence to the manuscript: "Indeed, the higher sensitivity of PA analysis based on MS-fingerprinting compared to NMR-based PA analysis was already reported and discussed previously¹³."

- 2. Line 94. It is not clear why the authors started this investigation using partially acetylated chitosans with FA = 0.32 deriving from heterogeneous/homogeneous deacetylation or re-N-acetylation.**

We added the following sentence for clarification: "We chose chitosans with an intermediate FA of 0.32 because low FA samples may not possess enough A-units to identify patterns, and high FA HTDA samples are neither common nor easily available. Also, such chitosans can be depolymerized by both chitinases and chitinases, but to a limited extent only, so that intermediate size oligomers are produced."

- 3. Line 104: The authors introduce the parameter “triad strength” to obtain information about DP = 3 products after digestion. Why was the peak at DP = 9 excluded in the calculation of this parameter? While DP = 12 is almost the same for all investigated conditions and therefore would not play a significant role in the calculation, this would be the case for DP = 9, as it is significantly lower for the heterogeneously deacetylated situation (Figure 1a).**

The idea of the parameter triad strength was to establish a value that a) reflects the differences we observe in the RI-chromatograms and b) is reliable to determine for all samples. Concerning a): We have checked whether the triad strength calculated with integrals of DP 2-6 is suitable to express the observed differences and found this to be true. Concerning b): Especially for low FA samples, AaChio with its specificity to cleave after the motif “DA” only produces low amounts of oligomeric products. Because smaller oligomers are better detectable by MS and also separated with higher resolution by SEC, identifying individual RI peaks is easier for small oligomers and becomes harder with increasing DP. Moreover, it was easier to draw a comparable baseline for all samples when using the integrals of DP 2-6 compared to 2-9 or 2-12.

- 4. The authors must provide limitations of the triad strength parameter. For instance, taking a look at figure 2, it emerges that the parameter fails to be valid for chitosans showing a fraction of acetylated units in the range 0.4 - 0.6. This is particularly important, since chitosans showing this FAs are well-known to start to become soluble at neutral pH. Thus, in principle, a direct correlation between pattern (determined by triad strength/composition) and solubility can not provided for this set of chitosans. Thus, FA still would remain the main parameter to predict the solubility of chitosans at neutral pH.**

The reviewer raises a valid point here (and also in remark 1 above) when asking about the sensitivity and the limitations of the parameter triad strength to detect patterns at different FAs. We now performed additional *in silico* simulations to model the triad strength for chitosans of different FA and different strengths of the $DDA_{DP/3}$ pattern (new Supplementary Fig. S16) and included a whole new paragraph to the discussion. Briefly, the additional data show that it is less pronounced but still possible to detect A-unit overrepresentation at every third unit also for high FA chitosans by the parameter triad strength, but we simply did not detect it in our high FA HTDA samples. The formation of the $DDA_{DP/3}$ pattern starts only after considerable deacetylation already occurred, supporting the hypothesis that pattern formation is linked to secondary structure formation which is in turn linked to FA-dependent chitosan solubility as indicated by the reviewer.

- 5. Nitrous acid depolymerization should also be considered as control to validate the experiments reported in Figure 3, as it randomly cleaves the chitosan backbone in correspondence with the D-units, generating reactive 2,5-anhydro-D-mannose residues that can subsequently be reduced.**

As suggested, we have now performed a nitrous acid depolymerization of the three differently produced FA 0.32 samples as well as of the HTDA and CNA sample with similar DP and FA 0.17. We added the RI signals to Figure 3 and Supplementary Figure S7, the product profiles to Supplementary Figure S6 and the efficiency of the cleavage to Supplementary Table 2, and included the results in corresponding paragraphs of the

manuscript. Unexpectedly, we observed “3n-steps” in the chromatograms of the products, but again only in the heterogeneously deacetylated chitosans. When we modeled the reaction *in silico*, the same steps were visible! Thus, nitrous acid depolymerization indeed added further and independent evidence for the described pattern. We refer to this new data (including Supplementary Fig. S14) to the last part of the results section where we had already mentioned nitrous acid deamination (but falsely assuming that its products would not be suitable indicators of the DDA_{DP/3} PA). We are grateful for the suggestion of the reviewer who encouraged us to perform these additional experiments.

6. Figure 3f. It is not clear why trifluoroacetic-assisted depolymerization would generate less products with DP = 3n in the case of heterogeneous deacetylation. This is in contrast with enzyme-assisted depolymerization.

In general, we observed differences in the degradability of HTDA, HMDA, and CNA substrates of FA 0.32, that are shown in Supplementary Table S2 and addressed in the corresponding part of the manuscript. We cannot yet answer the question why some enzymes (or TFA/nitrous acid) degrade chitosans of a certain production with higher or lower efficiency than others, but we will investigate this in future studies. Still, the take home message of Figure 3 lies in the differences of DP distributions between differently produced substrates for each enzyme, for which the different degradability is not of importance.

7. Bioactivity experiments. The authors claim that chitosans (and their products) from different chemical procedure elicit a different oxidative burst using *A. thaliana* as a model. This is interesting. The authors must exclude that differences in terms of oxidative stress would be related to endotoxins present in chitosan samples rather than their chemical composition. LAL tests on chitosan samples could be useful to test this.

The potential presence of endotoxins in chitosan samples is a recurrent and valid concern especially in bioactivity studies involving human/animal cells which can be extremely sensitive to endotoxins. Therefore, when we collaborated with colleagues in medicine, they repeatedly tested endotoxin levels in HTDA chitosans originating from Gillet Chitosan and in CNA chitosans prepared in our lab – and never found any. Still, we have now performed the suggested endotoxin test and added the results to the manuscript. We found levels of endotoxins below 0.4 EU/mL for all samples at the highest concentration tested in eliciting (100 µg/mL) (Supplementary Fig. S9). This is even below the FDA limit recommended for eluates from medical devices (≤ 0.5 EU/mL), and strong eliciting was also observed for HTDA samples of 1 µg/mL, which is again a 1:100 dilution. More importantly, the highly eliciting active HTDA samples do not contain more endotoxins than less eliciting active samples, indicating that indeed the differences between the chitosans, and not differences in endotoxin levels, are responsible for the different eliciting activities.

In addition, we are currently performing more bioassays with a broader set of differently produced chitosans and using different target cells/organisms for a follow-up publication. So far, all of them are confirming the trend that HTDA chitosans are more elicitor-active than CNA chitosans with random PA (see also remark 8). We now added further oxidative burst results to the supplements (Supplementary Fig. S10) and to the corresponding part

of the manuscript. These compare a different set of chitosan samples, one HTDA chitosan and two CNA chitosans with comparable FAs, concerning their elicitation activity in potato leaf discs.

- 8. Experiments on bioactivity. It should be very interesting to carry out further experiments on bioactivity and compare chitosans from different chemical processes. What about oxidative stress in immune cells or antibacterial properties, for example?**

We fully agree with the reviewer that (apart from looking into the molecular basis of pattern formation) the next step is to determine how relevant this pattern is for the chitosans' different bioactivities and for its performance in applications. To follow up on the study presented here, we have initiated corresponding experiments, including immune assays and antibacterial tests.

Reviewer 2

The manuscript is a very strong contribution to the important topics of chitosan synthesis, analysis, and bioactivity. It is also a strong contribution to the very knotty problem of determining and controlling sequence of monosaccharides in polysaccharides. Polysaccharides are one of a few great families of natural polymers, alongside proteins and poly(nucleic acid). The tools for understanding their structures are much less powerful than those in the hands of protein and DNA/RNA investigators, due in large part to the far, far greater complexity of polysaccharide structure. The manuscript by Moerschbacher et al. is an important contribution with potential impact across a number of scientific fields (e.g. chemistry, biology, agriculture, polymer science).

It is very well written, methodology is inventive and appropriate, and it is logically presented. I really only have two suggestions for improvement. Very strong, novel, impactful piece of work.

We are grateful for the very positive feedback of the reviewer and have addressed her or his valid remarks as indicated below, including amends to the manuscript that are indicated in the Word file of the revised manuscript using tracked changes.

- 1. The statement on page 2 about chitosans being rare in nature is not really accurate. It certainly is true that there are not nearly as many cationic polysaccharides as there are neutral and anionic ones. However, chitosan is a common element of fungal cell walls, and fungi are quite abundant. Thus, this statement should be revised or removed.**

It is difficult to reconcile this suggestion with that of referee 3 (point 2) who argues that chitosan does not really occur in nature. In fact, we tend to disagree with both referees. Partially deacetylated chitins which are soluble in acidic solution can be extracted from some fungal cell walls, but – with rare exceptions – from Mucoromycetes fungi only¹⁴⁻²⁰. We have revised our text in a way that we hope will satisfy both reviewers and removed the explicit statement that “chitosans are rare in nature”.

- 2. The structural arguments made about chitosan sequence following hydrolysis are complex, and I suspect will be difficult for some non-experts to follow. Especially since this is a journal with a broad scientific readership, it would be best if the arguments are presented with as high clarity as possible. I would suggest some use of structural cartoons, and of simple depictions of some of the proposed mechanisms by which regular patterns of acetylation are achieved, according to the hypotheses of the authors.**

As recommended by the reviewer, we included structural cartoons to the manuscript (Fig. 6) that visualize some of the proposed mechanisms to make our hypotheses more understandable for a wider audience.

Reviewer 3

This is a nice paper that reaches an important conclusion that will be of interest to researchers working on the production and application of chitosan. In recent years, the Moerschbacher team has made major steps that are needed to “finally” be able to properly characterize chitosans and, thus, to connect chitosan properties to biological function. This is crucial in a field that is big but that, for long, has suffered from mediocre publishing. The prime observation, non-random acetylation of heterogeneously deacetylated chitosans, is important, and well documented by studies with multiple enzymes, multiple chitosans and modelling. The fact that the authors present data for different chitosans and enzymes is laudable and adds reliability and impact. The paper is well written and easy to follow, despite the rather specialized nature of the topic. I have a few concerns and some suggestions for improvement, and I point at some issues that require clarification. Some of my suggestions are meant to make this paper more accessible to readers who are less familiar with the field.

We appreciate the very positive evaluation and are grateful for the extensive and helpful suggestions that we have addressed as indicated below, including amends to the manuscript that are indicated in the Word file of the revised manuscript using tracked changes.

1. **Line 17: “and found that both assumptions are wrong.” This is confusing (because the reader may associate this with assumptions regarding chitosans produced by homogeneous deacetylation or chemical N-acetylation, which is not what the authors intend to say); please rephrase.**

We have rephrased this part accordingly.

2. **Line 25 & 38 & 88-90: I my view, one should not give the impression that chitosan occurs in nature and I would advise against using the term “enzymatically produced natural chitosans” in Line 25. Yes, partially deacetylated chitin occurs in nature, but chitosans, as I see these, so acid-soluble polymeric materials in which all chains (not only the surface of a fibril) are deacetylated, and that give viscous solutions, do not really occur in nature. It is not possible to dissolve chitin by treating with deacetylases.**

As discussed above in response to point 2 of reviewer 2, it is difficult to reconcile the views of the two reviewers. Both referees seem to agree that our statement concerning chitosans being “rare in nature” is wrong, but while reviewer 2 says, it is abundant, reviewer 3 says, it does not exist. As mentioned, we tend to disagree with both referees. Partially deacetylated chitins which are soluble in acidic solution can be extracted from some fungal cell walls, but – with rare exceptions – from Mucoromycetes fungi only¹⁵ (such as *Mucor rouxii*¹⁷⁻²⁰, *Absidia* sp.^{14,17}, *Cunninghamella* sp.¹⁴, or *Rhizopus oryzae*^{14,17}), even using only acetic acid¹⁸. We have revised our text in a way that we hope will satisfy both reviewers and removed the explicit statement that “chitosans are rare in nature” as well as the term “natural chitosans”. We do agree with the reviewer that chitin deacetylases appear to be unable to deacetylate crystalline chitin, so we have added the clarification that enzymatic deacetylation *in vivo* most likely occurs on the nascent chitin chains emanating from the transmembrane chitin synthase before crystallization and fiber formation occurs.

- a. **Line 38 “Chitosans are rare in nature”. Do they or do they not occur? And if yes, how then do the authors define “chitosan”? A high quality, reliable reference is needed here.**

Please see our response to the point above.

(We follow the definition of “chitosan” as suggested by the European Chitin Society, namely as linear co-polymers of β -1,4-linked GlcN and GlcNAc units which are soluble in dilute acid. We agree with the reviewer that “high quality, reliable references” for the occurrence of chitosan in fungal cell walls are hard to find. Currently, we have to rely on a number of studies of different quality which in their entirety, to us at least, seem to suggest that chitosans do indeed occur naturally, though rarely. We are currently using our new and more sophisticated analytical tools such as described here²¹, hoping to soon be ready to contribute a reliable publication to this controversially discussed field.)

- b. **Line 88 & 90: “Our data also suggest that commercial chitosans fundamentally differ in their structure from natural chitosans produced by the action of chitin deacetylases, which are highly unlikely to generate the same, regular PA.” This text must be changed for multiple reasons: (1) do natural chitosans even exist? (2) Deacetylases can deacetylate chitin but cannot convert chitin to chitosan, & (3) if they could there is no basis whatsoever to state that they are “highly unlikely to generate the same, regular PA” (the contrary is true; enzymes can do exactly this, in principle).**

- (1): see above
- (2): see also above. It is true that chitin deacetylases (CDAs) cannot act on crystalline chitin, but for example for *Cryptococcus neoformans*, GPI-anchored CDAs are reported to act on nascent chitin chains right after their synthesis by chitin synthases *in vivo*²², before the chitin molecules form crystalline structures. In general, deacetylation of chitin in cell walls of certain fungal species is believed to be the result of the action of CDAs which – we explicitly agree! – may create certain patterns in the resulting chitosan based on their specificities or preferences.
- (3): We agree, that creating and controlling patterns is exactly what enzymes can do and this is one of the main reasons our working group focuses on chitosan related enzymes. Before we found the more regular DDA_{DP/3} pattern, we even believed that CDAs are (at least so far) the only option to introduce regular/controlled patterns into chitosan polymers. But even though CDAs are hypothetically able to create distinct PAs when acting on nascent chitin chains *in vivo*, we think it is highly unlikely that from all the possible patterns, exactly the same regular DDA_{DP/3} pattern would be the one that is produced in fungal cell wall chitosan by CDAs. We emphasized this now: “It is highly unlikely, that these enzymes generate the **exact** same, regular PA as generated by HTDA.”

Please make related corrections throughout the manuscript.

3. **Line 42-45: The use of the terms heterogeneous and homogeneous is jargon. By just reading the text as is, the two terms are confusing, for example considering the meaning of the term “homogeneous catalysis”. I invite the authors to use a few**

more words to explain what is meant by these two terms and, importantly, to explain how exactly the two processes differ (they both seem “heterogeneous” to me). This is of particular importance since one key point of this paper is that the process used has impact on the pattern of deacetylation. (Nb. See also remark at Lines 141-143/332-334, below).

The rightly reviewer raises an interesting question. The terms homogeneous and heterogeneous have been used to distinguish between the two types of deacetylation reaction ever since “homogeneous” deacetylation was first reported, and they have been used faithfully ever since, including by ourselves, without questioning them. But of course, the reviewer is right: both types of reaction are, in truth, heterogeneous. We discussed this issue with Prof. Martin Peter, one of the most experienced chitin and chitosan chemists worldwide. We finally decided to keep the terms because they are well introduced in the community, but to follow the advice of the reviewer and include a few more words to clarify the differences between heterogeneous and homogeneous deacetylation. Moreover, we addressed the fact that all chitin deacetylations are heterogeneous reactions when using the correct definitions of heterogeneous reactions involving reactants in two or more phases, and homogeneous reactions involving reactants in a single phase only.

- 4. Line 72-74: Instead of, or next to, reference 38, the authors should cite some original work here, for example the pioneering work by Kjell Varum, who studied both GH18s and GH19s, some 15 years ago (e.g. Sorbotten 2005, Sasaki 2006, Horn 2006, Heggset 2009).**

Good point! We included the suggested references Sørbotten 2005²³ for GH18 and Heggset 2009²⁴ for GH19 chitinases, and also Sasaki 2006²⁵ which addresses both families.

- 5. Line 74-77: This was not easy to follow for this reviewer. I suggest some rephrasing and perhaps some more words to clarify the issues for the less informed reader. Two specific comments: (a) Please add a statement about the (lack of) specificity of AaChio in subsites +1 and +2. (b) I do not understand this sentence: “This enzyme therefore hydrolyzes chitosans precisely next to each transition from D- to A blocks”. What does this mean and how can this be inferred? What kind of “blocks” are we referring to here? And what kind of “transition”? This enzyme cuts after DA diads which, to me, does not seem linked to a “transition from D- to A blocks”. I recommend rephrasing and some expansion of lines 74-77.**

As suggested, we added a statement about the missing specificity at +1 and +2, and expanded the explanation how the products of AaChio are linked to and can give information on the length or size of A- and D-blocks of the polymer substrate. As this information has been described in some detail in our earlier paper where we first described this enzyme²⁶, we have added this explanation – which we agree to be helpful to the readers – to the supplement, and also added a cartoon to help understanding the complex issue (Supplementary Fig. S4).

- 6. Lines 141-143 & 332-334: Here the authors point at HMDA chitosans prepared using a “less homogeneous process” that show intermediate behavior. I struggle with this for two reasons**
 - a. What does “less homogeneous” mean? Reference? Where can I find the recipe for this “less homogeneous” process? What exactly is the difference?**

We agree that the term “less homogeneous” for Trondheim chitosans in comparison to Viscosans can be confusing and is, in fact, not correct (a reaction can be either homogeneous or heterogeneous, not something in between). Therefore, we decided to change the wording, using “homogeneous conditions” for Trondheim chitosans and citing the corresponding references, and “optimized homogeneous process” for Viscosans. Unfortunately, we cannot give information on the exact details of the Viscosan production process and how it differs from the original Trondheim process, as these are a company secret of Flexichem AB. But we have talked about this in detail with Dr. Mats Andersson, the CEO of Flexichem AB, who is familiar with both processes and stated that the process for the production of Viscosans is an optimized version of the Trondheim process.

- b. These Trondheim chitosans have been used to unravel processivity in chitinases (Varum, Eijsink et al., 2005 - 2010) and one of the key elements of data interpretation in those elegant studies was the assumption that these chitosans have random acetylation. Indeed, in those studies, there are no indications that the acetylation pattern is non-random (e.g., “intermediate”). Can these previous and the present observations be reconciled? Perhaps this should be discussed to some extent. Of note, these are just two chitosans of very many, so even deleting these would not really affect the paper or its conclusions.**

We are aware that these Trondheim chitosans were essential for various pioneering studies (such as Sørbotten 2005²⁰ and Horn 2006²³) and also our group has worked with similar materials produced by Kjell Vårums group (e.g. Vander 1998²⁴), assuming that their pattern is random. As we discussed in this manuscript, it is not easy to detect this special, more regular DDA_{DP/3} PA. Neither NMR-based methods nor DP distributions of small oligomers after nitrous acid deamination are suitable to efficiently unveil it, and it only becomes visible in DP distributions of products of hydrolyses with certain specificities, which is why we and, apparently, all others have overlooked it for years. Moreover, the Trondheim chitosans show only slightly increased triad strengths (especially for higher FAs), meaning that their PA is “closer to random” than for HTDA chitosans (as stated now in the discussion).

- 7. Lines 232-239: I appreciate the differences that are demonstrated, but I find the discussion a bit too firm. Yes, there are differences and they should be pointed out, but I think that text like “For the latter, the cleavage of D-rich regions by CsnMN releases eliciting active products with A-rich centers, whereas the destruction of immunomodulatory A-rich parts by TvChi drastically reduces the elicitation activity*.” goes too far. This is both unprecise and speculative. You cannot really connect the enzymes to “D-rich” and “A-rich” regions. What is an “A-rich center”? What is meant with “immunomodulatory A-rich parts”? Where is the evidence that these “parts” are “immunomodulatory”? What does immunomodulatory mean in this context? What does the * mean?**

Another concern is that TcChi reduces bioactivity a lot, while, as the author state “only limited hydrolysis occurred and a considerable amount of polymer remained.” How can this be explained?

The * refers to a footnote stating: “reference pending, manuscript submitted in parallel to this manuscript: Richter, C., Cord-Landwehr, S., Singh, R. & Moerschbacher, B. M. Dissecting and finetuning bioactivities of chitosans by enzymatic modification”. The manuscript in question should be available soon, and includes a detailed study of the elicitation activities of a well-characterized chitosan polymer and its enzymatic hydrolysates using CsnMN and TvChi. Moreover, we rephrased the corresponding paragraph to improve comprehensibility.

Concerning the (truly amazing and surprising) strong decrease of elicitation activity by only partial TvChi digestion, we believe that the strongly elicitation active motifs only represent a small part of the HTDA polymer sample and that these (more highly acetylated) patches are exactly the ones preferentially cleaved by TvChi. Other fractions of the sample remain intact, but also do not contain the elicitation active motifs. That way, even limited digestion can have a strong impact on bioactivity. We added a corresponding sentence to the manuscript.

8. Table S2: More details need to be provided: were the reactions complete? How did the authors ensure this? Add information about the cleavage specificity of the enzymes. Add data for AaChio.

For all AaChio digestions (also the ones forming the basis of the triad strength analyses), the reactions were complete. As we are currently expressing and characterizing this enzyme within the context of another study, we have detailed data on its activity, and ensured a full digestion by using sufficient amounts of enzyme and prolonged reaction times based on this data. We added this information to the materials and method part.

For all other enzymes, we have not ensured that the end point is reached but focused just on sufficient amounts of oligomer products to check for differences between the different substrates. The proof of principle that also other hydrolyses with certain preferences can unveil the $DDA_{DP/3}$ PA in HTDA chitosans does not require the end point to be reached. Moreover, we never compare efficiencies between enzymes but only between substrates in the manuscript when referring to Supplementary Table S2, and here reactions with the same enzyme were always performed simultaneously. But in case this reviewer is still critical of Supplementary Table S2 and the corresponding sentence in the manuscript, we would agree to remove them.

As requested, we added the subsite preferences of the enzymes and the corresponding preferred cleavages of the chemical reagents, as well as the data for AaChio (and also for nitrous acid deamination as requested by another reviewer).

9. Legend to Fig. S5. “The faster chemical hydrolysis after A-units48 is indicated accordingly.” I have no idea what this means. You are not measuring rates here. Reference 48 does not belong here.

The sentence is preceded by the explanation of the subsite preference nomenclature for enzymes (e.g. DD|XX or aA|Xa) and is supposed to explain that the preferential chemical acid hydrolysis of glycosidic bonds after A-units by TFA is shown using the same nomenclature (Xa|XX). The cited reference Vårum 2001²⁹ reports the corresponding faster acidic hydrolysis of A-A and A-D bonds in comparison with D-D and D-A bonds in concentrated HCl.

We rephrased the sentence in all corresponding captions to improve comprehensibility and added a second reference (Einbu 2007³⁰).

Other issues

- 1. In the main text and in the supplementary material, there are several Figures and Tables showing data for chitosan degradation by different enzymes (Figs. 3, S6, S7; Table S2). I would have preferred if results for the AaChio were included in these Figures and Tables for reference. I realize that these results are shown elsewhere, but it is not so easy to compare.**

As suggested, we have added AaChio products (and also nitrous acid deamination products as requested by another reviewer) to Figure 3, Figure S6, Figure S7 and Table S2.

- 2. In the manuscript, references are made to Kjell Varum and the group of Kjell Varum. Unfortunately, Kjell Varum died in 2017 and there is no longer such a thing as the group of Kjell Varum. I think that this somehow needs to be made clear in the paper. Please consider some rephrasing.**

We changed “group of Kjell Vårum” to “former group of Kjell Vårum” throughout the manuscript and indicated in materials and methods, that we have received these Trondheim samples already in 2003.

- 3. PA – please decide if the A is in subscript or not and check the complete manuscript (e.g., line 61).**

Our idea was to abbreviate “pattern of acetylation” as “PA”, but to use “P_A” when talking about the “P_A value” ranging from 0 (perfectly block-wise) through 1 (perfectly random) to 2 (perfectly alternating) as introduced by Weinhold et al.³¹ But we see how this may be confusing and changed “P_A value” to “PA value”, so the A is consistently not in subscript.

- 4. Line 219: rephrase. Which tendency?**

We rephrased the sentence to clarify: “The HTDA polymer triggered the strongest oxidative burst and caused elicitation at low concentrations whereas the CNA polymer and especially the HMDA polymer had a weaker effect and higher concentrations were required for efficient elicitation (Fig. 4a).”

- 5. Fig. 4b,c: please show cleavage specificities on the two enzymes, similar to what is done in Fig. 3.**

As suggested, we added the subsite preferences to Figure 4b+c, and in addition to Supplementary Figure S8.

- 6. Line 248: A reference must be provided for the TFA cleavage specificity.**

We added requested citations on the acid catalyzed hydrolysis of chitosan and its specificity in terms of different bonds (Vårum 2001²⁹, Einbu 2007³⁰).

References

1. de Souza, J. R. & Giudici, R. Effect of diffusional limitations on the kinetics of deacetylation of chitin/chitosan. *Carbohydr. Polym.* **254**, 117278; doi:10.1016/j.carbpol.2020.117278 (2021).
2. Ottøy, M. H., Vårum, K. M. & Smidsrød, O. Compositional heterogeneity of heterogeneously deacetylated chitosans. *Carbohydr. Polym.* **29**, 17–24; doi:10.1016/0144-8617(95)00154-9 (1996).
3. Roberts, G. A. F. Chitosan production routes and their role in determining the structure and properties of the product. in *Advances in Chitin Science Volume II* (eds. Domard, A., Roberts, G. A. F. & Vårum, K. M.) 22–31; (Jaques André, 1997).
4. Chen, C. H., Wang, F. Y. & Ou, Z. P. Deacetylation of β -chitin. I. Influence of the deacetylation conditions. *J. Appl. Polym. Sci.* **93**, 2416–2422; doi:10.1002/app.20753 (2004).
5. Lamarque, G., Viton, C. & Domard, A. Comparative Study of the First Heterogeneous Deacetylation of α - and β -Chitins in a Multistep Process. *Biomacromolecules* **5**, 992–1001; doi:10.1021/bm049780k (2004).
6. Lamarque, G., Viton, C. & Domard, A. Comparative Study of the Second and Third Heterogeneous Deacetylations of α - and β -Chitins in a Multistep Process. *Biomacromolecules* **5**, 1899–1907; doi:10.1021/bm049780k (2004).
7. Jiang, C. J. & Xu, M. Q. Kinetics of heterogeneous deacetylation of β -chitin. *Chem. Eng. Technol.* **29**, 511–516; doi:10.1002/ceat.200500293 (2006).
8. Lamarque, G., Chaussard, G. & Domard, A. Thermodynamic Aspects of the Heterogeneous Deacetylation of β -Chitin: Reaction Mechanisms. *Biomacromolecules* **8**, 1942–1950; doi:10.1021/bm070021m (2007).
9. Bradić, B., Bajec, D., Pohar, A., Novak, U. & Likozar, B. A reaction–diffusion kinetic model for the heterogeneous N-deacetylation step in chitin material conversion to chitosan in catalytic alkaline solutions. *React. Chem. Eng.* **3**, 920–929; doi:10.1039/C8RE00170G (2018).
10. Sannan, T., Kurita, K. & Iwakura, Y. Studies on Chitin, 1: Solubility Change by Alkaline Treatment and Film Casting. *Die Makromol. Chemie* **176**, 1191–1195; (1975).
11. Sannan, T., Kurita, K. & Iwakura, Y. Studies on Chitin, 2: Effect of Deacetylation on Solubility. *Die Makromol. Chemie* **177**, 3589–3600; doi:10.1271/nogeikagaku1924.23.437 (1976).
12. Kurita, K., Sannan, T. & Iwakura, Y. Studies on Chitin, 4. Evidence for Formation of Block and Random Copolymers of N-Acetyl-D-glucosamine and D-Glucosamine by Hetero- and Homogeneous Hydrolyses. *Die Makromol. Chemie* **178**, 3197–3202; doi:https://doi.org/10.1002/macp.1977.021781203 (1977).
13. Wattjes, J. *et al.* Enzymatic Production and Enzymatic-Mass Spectrometric Fingerprinting Analysis of Chitosan Polymers with Different Nonrandom Patterns of Acetylation. *J. Am. Chem. Soc.* **141**, 3137–3145; doi:10.1021/jacs.8b12561 (2019).
14. Vaingankar, P. N. & Juvekar, A. R. Fermentative Production of Mycelial Chitosan from Zygomycetes: Media Optimization and Physico-Chemical Characterization. *Adv. Biosci. Biotechnol.* **05**, 940–956; doi:10.4236/abb.2014.512108 (2014).
15. Ghormade, V., Pathan, E. K. & Deshpande, M. V. Can fungi compete with marine sources for chitosan production? *Int. J. Biol. Macromol.* **104**, 1415–1421; (2017).
16. Lecoïnte, K., Cornu, M., Leroy, J., Coulon, P. & Sendid, B. Polysaccharides Cell Wall Architecture of Mucorales. *Front. Microbiol.* **10**, 1–8; doi:10.3389/fmicb.2019.00469 (2019).
17. Hu, K.-J., Hu, J.-L., Ho, K.-P. & Yeung, K.-W. Screening of fungi for chitosan producers, and copper adsorption capacity of fungal chitosan and chitosanaceous materials. *Carbohydr. Polym.* **58**, 45–52; doi:10.1016/j.carbpol.2004.06.015 (2004).
18. Synowiecki, J. & Al-Khateeb, N. A. A. Q. Mycelia of *Mucor rouxii* as a source of chitin and chitosan. *Food Chem.* **60**, 605–610; doi:10.1016/S0308-8146(97)00039-3 (1997).
19. Arcidiacono, S. & Kaplan, D. L. Molecular weight distribution of chitosan isolated from *Mucor rouxii* under different culture and processing conditions. *Biotechnol. Bioeng.* **39**, 281–286; doi:10.1002/bit.260390305 (1992).

20. Wu, T., Zivanovic, S., Draughon, F. A., Conway, W. S. & Sams, C. E. Physicochemical Properties and Bioactivity of Fungal Chitin and Chitosan. *J. Agric. Food Chem.* **53**, 3888–3894; doi:10.1021/jf048202s (2005).
21. Urs, M. J., Moerschbacher, B. M. & Cord-Landwehr, S. Quantitative enzymatic-mass spectrometric analysis of the chitinous polymers in fungal cell walls. *Carbohydr. Polym.* **301**, 120304; doi:10.1016/j.carbpol.2022.120304 (2023).
22. Baker, L. G., Specht, C. A., Donlin, M. J. & Lodge, J. K. Chitosan, the deacetylated form of chitin, is necessary for cell wall integrity in *Cryptococcus neoformans*. *Eukaryot. Cell* **6**, 855–867; doi:10.1128/EC.00399-06 (2007).
23. Sørbotten, A., Horn, S. J., Eijsink, V. G. H. & Vårum, K. M. Degradation of chitosans with chitinase B from *Serratia marcescens*. *FEBS J.* **272**, 538–549; doi:10.1111/j.1742-4658.2004.04495.x (2005).
24. Heggset, E. B., Hoell, I. A., Kristoffersen, M., Eijsink, V. G. H. & Vårum, K. M. Degradation of chitosans with chitinase G from *Streptomyces coelicolor* A3(2): Production of chito-oligosaccharides and insight into subsite specificities. *Biomacromolecules* **10**, 892–899; doi:10.1021/bm801418p (2009).
25. Sasaki, C., Vårum, K. M., Itoh, Y., Tamoi, M. & Fukamizo, T. Rice chitinases: sugar recognition specificities of the individual subsites. *Glycobiology* **16**, 1242–1250; doi:10.1093/glycob/cwl043 (2006).
26. Kohlhoff, M. *et al.* Chitinase: A fungal chitosan hydrolyzing enzyme with a new and unusually specific cleavage pattern. *Carbohydr. Polym.* **174**, 1121–1128; doi:10.1016/j.carbpol.2017.07.001 (2017).
27. Horn, S. J. *et al.* Endo/exo mechanism and processivity of family 18 chitinases produced by *Serratia marcescens*. *FEBS J.* **273**, 491–503; doi:10.1111/j.1742-4658.2005.05079.x (2006).
28. Vander, P., Vårum, K. M., Domard, A., El Gueddari, N. E. & Moerschbacher, B. M. Comparison of the ability of partially n-acetylated chitosans and chito-oligosaccharides to elicit resistance reactions in wheat leaves. *Plant Physiol.* **118**, 1353–1359; doi:10.1104/pp.118.4.1353 (1998).
29. Vårum, K. M., Ottøy, M. H. & Smidsrød, O. Acid hydrolysis of chitosans. *Carbohydr. Polym.* **46**, 89–98; doi:10.1016/S0144-8617(00)00288-5 (2001).
30. Einbu, A., Grasdalen, H. & Vårum, K. M. Kinetics of hydrolysis of chitin/chitosan oligomers in concentrated hydrochloric acid. *Carbohydr. Res.* **342**, 1055–1062; doi:10.1016/j.carres.2007.02.022 (2007).
31. Weinhold, M. X., Sauvageau, J. C. M., Kumirska, J. & Thöming, J. Studies on acetylation patterns of different chitosan preparations. *Carbohydr. Polym.* **78**, 678–684; doi:10.1016/j.carbpol.2009.06.001 (2009).

Reviewers' Comments:

Reviewer #1:

Remarks to the Author:

The authors have responded point-by-point to the comments I made in the last round of revisions. In general, I am quite satisfied with their answers. As mentioned earlier, I felt that the molecular basis for this unexpected regular recurrence of PA in heterogeneously deacetylated chitosans needs to be demonstrated in this manuscript to make a fundamental breakthrough in this field. Now, in my opinion, the authors have provided a solid rationale to my concerns, and I agree it is not easy to verify such aspect. I have appreciated their efforts to explain a possible model for such unexpected results. This was achieved after discussions with experts in the field and presented in the revised manuscript (scheme in Figure 6). Overall, I believe that the authors have now found a good compromise in the discussion of their results. The second (critical) point was to perform additional experiments to investigate the bioactivity of these chitosans from different chemical processing. Although they provided additional results on the oxidative burst, I am still convinced that more experiments need to be included in this work to strengthen the scientific message of this manuscript. As the authors are currently conducting antibacterial experiments, it would be very useful to see a proof of principle in this manuscript regarding antimicrobial activity. The same chitosans that were used in the experiments with *A. thaliana* should be used for this purpose.

Reviewer #2:

Remarks to the Author:

The authors have done a good job of addressing prior concerns, and the manuscript can be published in its current form, from my perspective. A few small suggestions to improve the final version:

L52 this sentence is a bit awkward as written. Perhaps something like "Because beta-chitin has weaker self-association than alpha-chitin..."

Figure 2: Nice figure but as presented, can only be understood in color. Readers who are color-blind or who have printed in black and white could not understand. Better to use datapoints that differ in both color and shape by series.

L194, L240 negligible

L240 ruling out the possibility that...

L260 lightly acetylated...

Reviewer #3:

Remarks to the Author:

The authors have responded well to my comments and the comments provided by the other reviewers. I am very happy with the way they dealt with my comments. Among other things, the changes made in the Introduction should make the paper more accessible for the less-informed reader. I like the way in which the authors have dealt with the comments related to natural chitosan and homogeneous vs heterogeneous. The changes made in the section about elicitation are also fine. I liked the expanded speculation about the mechanism behind the non-randomness of deacetylation (an issue raised by one of the other reviewers).

I have no further comments, but would like to note that I found it a bit hard to follow the newly added text near Fig. 3 that discusses the (nice) new data obtained with nitrous acid. In particular, the term

"3n-steps" confused me. Perhaps rephrase a bit and/or add a reference to the explanatory text that appears towards the end of the Results section.

Point-by-point response to reviewers

Reviewer 1

The authors have responded point-by-point to the comments I made in the last round of revisions. In general, I am quite satisfied with their answers. As mentioned earlier, I felt that the molecular basis for this unexpected regular recurrence of PA in heterogeneously deacetylated chitosans needs to be demonstrated in this manuscript to make a fundamental breakthrough in this field. Now, in my opinion, the authors have provided a solid rationale to my concerns, and I agree it is not easy to verify such aspect. I have appreciated their efforts to explain a possible model for such unexpected results. This was achieved after discussions with experts in the field and presented in the revised manuscript (scheme in Figure 6). Overall, I believe that the authors have now found a good compromise in the discussion of their results. The second (critical) point was to perform additional experiments to investigate the bioactivity of these chitosans from different chemical processing. Although they provided additional results on the oxidative burst, I am still convinced that more experiments need to be included in this work to strengthen the scientific message of this manuscript. As the authors are currently conducting antibacterial experiments, it would be very useful to see a proof of principle in this manuscript regarding antimicrobial activity. The same chitosans that were used in the experiments with *A. thaliana* should be used for this purpose.

We are glad to hear that we have met the reviewer's concerns about the molecular basis of the regular PA, and want to thank her or him for making us clarify our hypotheses with an additional schematic figure. Moreover, we agree that investigating the consequences of our observation of the regular PA in HTDA chitosans for their many applications is a logical and important next step. But to do so properly will require us to include a larger set of chitosan samples with similar DP and FA but prepared in different ways so that they differ in PA, performing a sufficient number of independent replications for adequate statistics, and that for a sizeable range of bioactivities towards different cells and organisms. These studies are under way and will hopefully be ready for submission later this year, but this will depend on the progress of the work which we are performing in collaborations with experts in different fields. The results on plant disease resistance eliciting activities which we included in the current manuscript (now, triggered by the comment of the reviewer, corroborated using a second species) serve to clearly indicate that the structural difference we observe are of crucial relevance for the functional activities of the chitosans. We believe that adding preliminary data on antimicrobial activities - and at this point in time, only preliminary data would be available - will not really add to this message.

Reviewer 2

The authors have done a good job of addressing prior concerns, and the manuscript can be published in its current form, from my perspective. A few small suggestions to improve the final version.

We want to thank the reviewer for the overall very helpful feedback and have again changed the manuscript according to her or his final small suggestions.

1. **L52 this sentence is a bit awkward as written. Perhaps something like "Because beta-chitin has weaker self-association than alpha-chitin..."**

We changed the sentence to: "Because β -chitin exhibits weaker self-association through intermolecular interactions than α -chitin, ..."

2. **Figure 2: Nice figure but as presented, can only be understood in color. Readers who are color-blind or who have printed in black and white could not understand. Better to use datapoints that differ in both color and shape by series.**

Done.

3. **L194, L240 negligible**

Done.

4. **L240 ruling out the possibility that...**

Done.

5. **L260 lightly acetylated...**

Done.

Reviewer 3

The authors have responded well to my comments and the comments provided by the other reviewers. I am very happy with the way they dealt with my comments. Among other things, the changes made in the Introduction should make the paper more accessible for the less-informed reader. I like the way in which the authors have dealt with the comments related to natural chitosan and homogeneous vs heterogeneous. The changes made in the section about elicitation are also fine. I liked the expanded speculation about the mechanism behind the non-randomness of deacetylation (an issue raised by one of the other reviewers).

I have no further comments, but would like to note that I found it a bit hard to follow the newly added text near Fig. 3 that discusses the (nice) new data obtained with nitrous acid. In particular, the term "3n-steps" confused me. Perhaps rephrase a bit and/or add a reference to the explanatory text that appears towards the end of the Results section.

We appreciate all of the reviewer's helpful feedback during the whole revision process and have now rephrased the text around Figure 3 and at the end of the results section to make it easier to follow.